# JiangJun: Mastering Xiangqi by Tackling Non-Transitivity in Two-Player Zero-Sum Games

**Yang Li**[1,*]**, Kun Xiong**[2]**, Yingping Zhang**[2]**, Jiangcheng Zhu**[2]**, Stephen Mcaleer**[3]**, Wei Pan**[1]**, Jun Wang**[4]**, Zonghong Dai**[2,†]**, Yaodong Yang**[5,†]
[1] *The University of Manchester,* [2] *Huawei,* [3] *Carnegie Mellon University,* [4] *University College London,* [5] *Peking University*

**Reviewed on OpenReview:** *https://openreview.net/forum?id=MMsyqXIJuk*

## Abstract

This paper presents an empirical exploration of non-transitivity in perfect-information games, specifically focusing on Xiangqi, a traditional Chinese board game comparable in game-tree complexity to chess and shogi. By analyzing over 10,000 records of human Xiangqi play, we highlight the existence of both transitive and non-transitive elements within the game's strategic structure. To address non-transitivity, we introduce the JiangJun algorithm, an innovative combination of Monte-Carlo Tree Search (MCTS) and Policy Space Response Oracles (PSRO) designed to approximate a Nash equilibrium. We evaluate the algorithm empirically using a WeChat mini program and achieve a Master level with a 99.41% win rate against human players. The algorithm's effectiveness in overcoming non-transitivity is confirmed by a plethora of metrics, such as relative population performance and visualization results. Our project site is available at `https://sites.google.com/view/jiangjun-site/`.

## 1 Introduction

Multi-Agent Reinforcement Learning (MARL) has demonstrated remarkable success in various games such as Hide and Seek (Baker et al., 2019), Go (Silver et al., 2016b), StarCraft II (Vinyals et al., 2019), Dota 2 (Berner et al., 2019), and Stratego (Perolat et al., 2022). However, algorithms such as AlphaZero (Silver et al., 2018) and AlphaGo (Silver et al., 2016b) that train against the most recent opponent can potentially cycle in games that have non-transitive structure. While this problem has been well-studied in imperfect-information games (Lanctot et al., 2017; Perolat et al., 2022; McAleer et al., 2022a;c; 2021; Fu et al., 2022; Brown et al., 2019; Heinrich & Silver, 2016; Steinberger et al., 2020; Perolat et al., 2021; Hennes et al., 2019), it has been less studied in perfect-information games.

Conquering non-transitivity in perfect-information games remains an open research direction. Recent works (Balduzzi et al., 2019b; McAleer et al., 2020; Liu et al., 2021; McAleer et al., 2022c;b) have focused on finding (approximate) Nash equilibria using the Policy Space Response Oracles (PSRO) algorithm (Lanctot et al., 2017), which is developed from the Double Oracle (DO) algorithm (McMahan et al., 2003). These approaches deal with non-transitivity by mixing over a population of policies, but have to our knowledge not been studied in perfect-information games.

This study aims to investigate Xiangqi, a prevalent two-player zero-sum game with moderate game-tree complexity of $10^{150}$ (between Chess ($10^{128}$ game-tree complexity) and Go ($10^{360}$ game-tree complexity)), which has not received much attention in previous research. Although Xiangqi's complexity presents a challenge, it is not an insurmountable obstacle, and its accessibility makes it an excellent candidate for exploring the geometrical landscape of board games and non-transitivity.

In this study, we delve into the intricate geometry of Xiangqi, leveraging a dataset comprising over 10,000 game records from human gameplay as the foundational basis for our investigation. Our findings unveil the

---

*Part work done when Yang was an intern at Huawei.
†Corresponding authors: daizonghong@huawei.com, yaodong.yang@pku.edu.cn.

existence of a spinning top structure embedded within the game dynamics, and highlight the manifestation of non-transitivity within the mid-range ELO rating scores. By implementing self-play training on policy checkpoints, we further uncover strategic cycles that empirically substantiate the presence of non-transitivity in Xiangqi.

Motivated by these insights, we propose the JiangJun algorithm, specifically designed to mitigate the non-transitive challenge observed in Xiangqi. This algorithm incorporates two fundamental modules: the *MCTS actor* and *Populationer*. Together, these components approximate Nash equilibria within the player population, employing Monte Carlo Tree Search (MCTS) techniques. A thorough examination of the computational complexity associated with the JiangJun algorithm is also presented in this paper. Specifically, the worst-case time complexity of the MCTS actor is defined as $\mathcal{O}(b^d \times n \times C)$, where $b$, the effective branching factor of Xiangqi, falls within a median range of 30 to 80, the game tree depth $d$ extends beyond 40, the typical number of simulations $n$ is set to 800, and $C$ denotes the time required for neural network inference. Additionally, if the Simplex method is utilized, the Nash Solver corresponds to a worst-case time complexity of $O(2^k \times poly(k))$, with $poly(k)$ signifying a polynomial function in terms of $k$.

The efficacy of the JiangJun algorithm is thoroughly appraised via an expansive set of metrics in this study. The training of the JiangJun algorithm to the "Master" level was facilitated by our proposed training framework that effectively utilizes the computational capabilities of up to 90 V100 GPUs on the Huawei Cloud ModelArt platform. Initially, a diverse range of indicators, including relative population performance, Nash distribution visualization, and the low-dimensional gamescape visualization of the primary two embedding dimensions, collectively corroborate the proficiency of JiangJun in addressing the non-transitivity issue inherent in Xiangqi. In addition, JiangJun significantly surpasses its contemporary algorithms—standard AlphaZero Xiangqi and behavior clone Xiangqi—demonstrating winning probabilities exceeding 85% and 96.40% respectively. Furthermore, upon evaluation of exploitability, JiangJun (8.41% win rate of the approximate best response) was found to be appreciably closer to the optimal strategy in contrast to the standard AlphaZero Xiangqi algorithm (25.53%).

Additionally, we devised and implemented a Xiangqi mini-program on the WeChat platform, which, over a span of six months, compiled more than 7,000 game records from matches played between JiangJun and human opponents. The data generated from these matches yielded an extraordinary 99.41% win rate for JiangJun against human players post-training, thereby underscoring its formidable strength. Lastly, through an in-depth case study of various endgame scenarios, the capacity of JiangJun to astutely navigate the intricacies of the Xiangqi endgame is exhibited.

In summary, this paper presents three key contributions: 1) an analysis of the geometrical landscape of Xiangqi using over 10,000 real game records, uncovering a spinning top structure and non-transitivity in real-world games; 2) the proposal of the JiangJun algorithm to conquer non-transitivity and master Xiangqi; and 3) the empirical evaluation of JiangJun with human players using a WeChat mini program, achieving over a 99.41% win rate and demonstrating the efficient overcoming of non-transitivity as shown by other metrics such as relative population performance.

The remaining sections of this paper are organized as follows. Section 2 summarizes related work in the field. In Section 3, we give a brief introduction to the Xiangqi game. Section 4 presents our non-transitive analysis method and the results of real-world Xiangqi data analysis. Next, in Section 5, we introduce our proposed JiangJun algorithm, while Section 6 presents the experimental results. Finally, we draw our conclusions in Section 7. Additional details pertinent to this paper can be found in Appendix 7.

## 2  Related Work

**Opponen Selection Strategy in MARL.** MARL has achieved a huge breakthrough in the field of games such as StarCraft, DOTA, Majiang, and Doudizhu. To solve those complex games, many methods are based on self-play or training against previous strategies. As shown in Table 1, we provide an overview of opponent selection strategies used by 11 state-of-the-art algorithms, which we've grouped into 5 categories. The categories are 1) Latest: selects the most recent opponent strategy, such as AlphaZero (Silver et al., 2018), Suphx (Li et al., 2020), ACH (Fu et al., 2022), DouZero (Zha et al., 2021) and Perfect Dou (Guan et al.,

| Opponent Selection Strategy | Algorithms |
|---|---|
| Latest Strategy | AlphaZero(Silver et al., 2018),Suphx (Li et al., 2020), ACH (Fu et al., 2022), DouZero (Zha et al., 2021), PerfectDou (Guan et al., 2022) |
| History Random | AlphaGo (Silver et al., 2016b) |
| History Best | AlphaGo Zero(Silver et al., 2017b) |
| History Top K | AlphaHoldem (Zhao et al., 2022) |
| Population-Based | AlphaStar (Vinyals et al., 2019) |
| Mixed | OpenAI Five (Berner et al., 2019), Jue Wu (Wu et al., 2018) |

Table 1: The presented summary table provides an overview of the opponent selection strategies used by 11 leading algorithms, which are classified into six distinct categories: the latest strategy, random/best strategy in history, top-k strategies in history, population-based strategy, and a mixed sampling method that combines the latest and historical random strategies.

2022), 2) History Random: randomly samples a historical opponent's strategy like AlphaGo (Silver et al., 2016b), 3) History Best: selects the opponent strategy with the best performance like AlphaGo Zero (Silver et al., 2017b), 4) History Top K: selects the top-k historical opponent strategies like AlphaHoldem (Zhao et al., 2022), 5)Population-Based: select the opponent strategy from a population of strategies such as AlphaStar (Vinyals et al., 2019), and 6) Mixed category: combine the Latest and History Random strategies, with opponents selected according to a weighted combination of the two strategies (80% Latest and 20% History Random).

**Non-transitivity.** Recent research has examined the non-transitivity and corresponding reinforcement learning solutions. The spinning top hypothesis proposed by Czarnecki et al. (2020) describes the Game of Skill geometry, where the strategy space of real-world games consists of transitivity and non-transitivity, resembling a spinning top structure. The transitive strength gradually diminishes as game skill increases upward to the Nash Equilibrium or evolves downward to the worst possible strategies.

To address the non-transitivity in real-world games, especially zero-sum games, Double Oracle (DO) (McMahan et al., 2003)-based reinforcement learning methods have been proposed. These algorithms attempt to find the best response against a previous aggregated policy at pre-iteration. The DO algorithm is designed to derive Nash Equilibrium in normal-form games, which is guaranteed by finding a true best-response oracle(Yang et al., 2021).

For large-scale games, policy-space response oracles (PSRO) (Lanctot et al., 2017) naturally developed from the DO algorithm, finding an approximate best response instead of a true best response. In PSRO with two players, each player maintains a population $\mathcal{B}_{1,2} = \{\pi_{1,2}^1, \ldots, \pi_{1,2}^n\}$. When player $i \in \{1, 2\}$ tries to add a new policy $\pi_i^{n+1}$, the best response to the mixture of opponents will be obtained by following the equation.

$$\text{Br}(\mathcal{B}_{-i}) = \max_{\pi_i^{n+1}} \sum_j \sigma_{-i}^j E_{\pi_i^{n+1}, \pi_{-i}^j}[r_i(s, \mathbf{a})], \tag{1}$$

where $(\sigma_i, \sigma_{-i})$ is the Nash equilibrium distribution over policies in $\mathcal{B}_i$ and $\mathcal{B}_{-i}$. Variants of PSRO have been proposed, including PSRO with a Nash rectifier ($PSRO_{rN}$) (Balduzzi et al., 2019b) to explore strategies with positive Nash support to improve strategy strength, AlphaStar (Vinyals et al., 2019) for computing the best response against a mixture of opponents, and Pipeline PSRO (McAleer et al., 2020) for parallelizing PSRO by maintaining a hierarchical pipeline of reinforcement learning workers. Behavioral diversity is believed to be linked to non-transitivity, observed in both human societies and biological systems (Kerr et al., 2002; Reichenbach et al., 2007). As a result, some researchers have studied behavioral diversity to solve non-transitivity problems (Yang et al., 2021; Liu et al., 2021). However, the quantification and measurement of behavioral diversity are still not well understood (Yang et al., 2021). Some works define behavioral diversity as the variance in rewards (Lehman & Stanley, 2008; 2011), the convex hull of a gamescape (Czarnecki et al., 2020), and the discrepancies of occupancy measures (Liu et al., 2021).

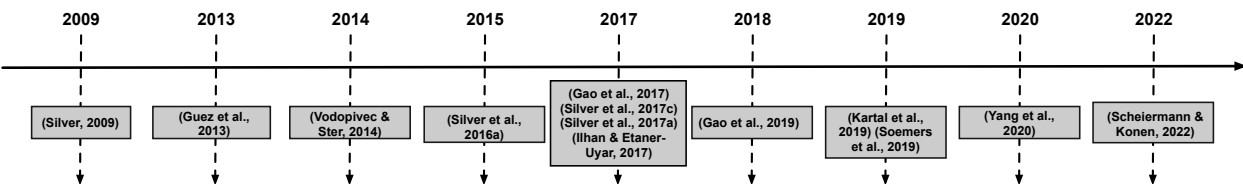

Figure 1: Timeline illustrating the evolution of the integration between Monte Carlo Tree Search (MCTS) and Reinforcement Learning (RL). The timeline is based on the release dates of the methods, rather than their paper publication dates.

While PPAD-Hard for computing Nash equilibrium (Balduzzi et al., 2019a), $\alpha-$PSRO (Muller et al., 2019) proposes a polynomial-time solution $\alpha-$Rank for general-sum games instead of Nash equilibrium. Other works focus on maximizing diversity in deriving the best response, including Diverse-PSRO (Liu et al., 2021), which offers a unified view for diversity in the PSRO framework by combining both the behavioural diversity and the response diversity of a strategy, and DPP-PSRO (Nieves et al., 2021), which combines a novel diversity metric, determinantal point processes, and best-response dynamics for solving normal-form and open-ended games. PSRO has also recently been applied to robust RL, where it can find policies that are robust to all feasible environments (Lanier et al., 2022).

**Monte Carlo Tree Search and Reinforcement Learning.** The integration of Monte Carlo Tree Search (MCTS) and Reinforcement Learning (RL) has been a significant area of exploration (Vodopivec et al., 2017; Swiechowski et al., 2022). The initial connection between MCTS and RL was established in Silver's 2009 PhD thesis (Silver, 2009). A scalable and efficient Bayes-Adaptive Reinforcement Learning method based on MCTS is introduced (Guez et al., 2013). This approach significantly outperformed previous Bayesian model-based reinforcement learning algorithms on several benchmark problems. The subsequent development of the TD-MCTS approach in 2014 (Vodopivec & Ster, 2014) marked a significant milestone, as it incorporated Temporal Difference (TD) learning into MCTS, thereby modifying the Upper Confidence Bound for Trees (UCT) formula and the state estimate calculations for nodes. The integration of MCTS and RL witnessed a breakthrough with the introduction of AlphaGo (Silver et al., 2016a) in 2015. This was followed by the application of the AlphaGo strategy to the game of Hex by MoHex-CNN (Gao et al., 2017) in 2017, and the development of AlphaGoZero (Silver et al., 2017c), which mastered Go without human knowledge. The same year also saw the advent of AlphaZero (Silver et al., 2017a), which excelled at chess, shogi, and Go, and the introduction of a technique by Ilhan et al. (Ilhan & Etaner-Uyar, 2017) that adjusted the policy during the MCTS simulation phase using the TD method. In 2018, MoHex-3HNN (Gao et al., 2019) significantly outperformed MoHex-CNN with its innovative three-head neural network architecture. The following year, Soemers et al. (Soemers et al., 2019) learned a policy in a Markov Decision Process (MDP) using the policy gradient method and value estimates directly from MCTS. MCTS was also used as a demonstrator for the RL component in (Kartal et al., 2019). More recently, in 2020, a method inspired by AlphaGo was developed to operate without prior knowledge of komi (Yang et al., 2020). In 2022, Scheiermann et al. integrated MCTS with TD n-tuple networks for the first time, using this combination only during testing to create adaptable agents while maintaining low computational demands (Scheiermann & Konen, 2022).

## 3 Xiangqi: the Game

Xiangqi, also known as Chinese chess, is an ancient and widely popular board game played worldwide. Its history dates back around 3,500 years to a game called Liubo. Over the centuries, the game evolved into the Xiangqi, played today, a two-player board game on a 9x10 board. The game's goal is for each player, red and black, to capture the other's king. The red player always plays first, and the two players alternate moves. Each player controls seven types of pieces for a total of sixteen pieces: one king (帅/将, K/k), two advisors (仕/士, A/a), two bishops (相/象, B/b), two rooks (車/車, R/r), two knights (馬/馬, N/n), two cannons (炮/炮, C/c), and five pawns (卒/兵, P/p). The first Chinese character or uppercase, represents pieces for the red player, and the second Chinese character or lowercase, represents pieces for the black player.

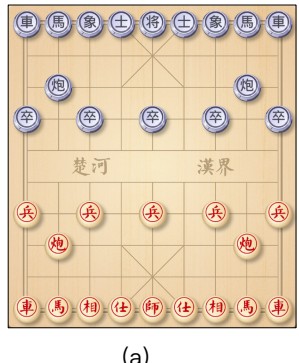

| Symbol | Piece | Chinese |
|--------|-------|---------|
| P/p | Pawns | 卒/兵 |
| C/c | Cannons | 炮/炮 |
| R/r | Rooks | 車/車 |
| K/k | Knights | 馬/馬 |
| B/b | Bishops | 相/象 |
| A/a | Advisors | 仕/士 |
| S/s | Kings | 帅/将 |

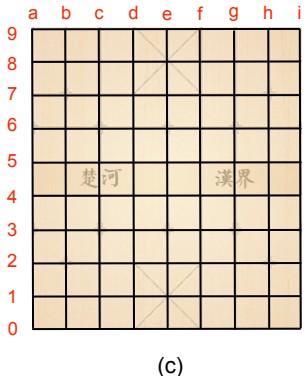

(a)       (b)       (c)

Figure 2: Introduction of Xiangqi. Left figure (a) shows the initial position of Xiangqi. To better describe Xiangqi, we use corresponding English characters (as shown in sub-figure (b)) and $9 \times 10$ (width×height) grid matrix (as shown in sub-figure (c)) to present pieces and Xiangqi board, respectively.

The game of Xiangqi is introduced in Figure 2, which provides the initial board setup and the corresponding English name and symbol. The game board is divided into two territories, red and black, by a horizontal line known as the "River". The Palace, marked with an "X" on each side, is located in the center bottom board and restricted to Kings and Advisors' movement only. Kings can move one step at a time in the horizontal or vertical direction, while Advisors are restricted to one-step diagonal movements within the Palace. If a King faces the other King, it will be captured, resulting in an automatic loss. The Bishops on either side of the Palace can only move diagonally within a $2 \times 2$ square on their respective side. Similarly, the Horses can move diagonally within a $1 \times 2$ rectangle. However, the movement of Bishops and Horses is blocked by other pieces. Bishops cannot move if a piece is positioned in the center of the $2 \times 2$ square, and a piece located along the long edge of a Horse prevents it from moving along the corresponding $1 \times 2$ rectangle. The Rooks and Cannons move freely in vertical or horizontal directions, with Cannons being capable of capturing other pieces by jumping over a single piece. Rooks, on the other hand, can only capture the first piece in their path. The game also features five Pawns on each side, which can move one space forward within their territory. Once they cross the River, they can also move one space forward, left, or right.

To provide a standardized way to describe the positions of the pieces in Xiangqi, we encode the Xiangqi game into a $9 \times 10$ matrix, as shown in Fig. 2 (c). The board matrix has been numbered using nine letters from $a$ to $i$ for the length and ten numbers from 0 to 9 for the width. This allows us to represent the current state and actions using a string of characters based on the Universal Chinese chess Protocol (UCCI). For example, the initial position shown in Fig. 2 (b) can be represented by the string "rnbakabnr/9/1c5c1/p1p1p1p1p/9/9/P1P1P1P1P/1C5C1/9/RNBAKABNR", where the numbers represent the number of empty spaces. The action of each piece can also be described using a tuple consisting of four characters, such as b0c2, which means that the piece moves from column b row 0 to column c row 2.

## 4    Real-World Xiangqi Data Analysis

In this section, we will present the analysis method and results of real-world Xiangqi data obtained from the Play OK game platform[1], a global gaming platform offering a variety of games to play with online players from around the world. Our analysis reveals that the geometric structure of the real-world Xiangqi game resembles that of a spinning top, in accordance with the Game of Skill hypothesis (Czarnecki et al., 2020). This finding suggests that non-transitivity exists in real-world Xiangqi gameplay.

---

[1]www.playok.com

### 4.1 Preliminaries for Measuring Non-Transitivity

To begin with, assuming that the probability $p$ of one player winning over another can be estimated or computed, the Xiangqi game can be represented by a tuple $(n, \mathcal{W}, \mathcal{M})$. Here, $\mathcal{W} = \{w_1, w_2, \cdots, w_n\}$ refers to the $n$ players or agents, who can be either human players or neural network-based agents. Furthermore, the payoff matrix $\mathcal{M} \in \mathbb{R}^{n \times n}$ is defined, where each element in $\mathcal{M}$ is calculated by a payoff function $\phi : w_i \times w_j \to \mathbb{R}$. The function $\phi$ is an antisymmetric function $\phi(w_i, w_j) = -\phi(w_j, w_i)$ for $i, j < n$ and $i \neq j$, and a higher value of $\phi(w_i, w_j)$ represents a better outcome for player $w_i$. The payoff function can be transformed into a win/loss probability via $\phi(w_i, w_j) := p(w_i, w_j) - 1/2$. Wins, losses, and ties for $w_i$ are represented by $\phi(w_i, w_j) > 0, \phi(w_i, w_j) < 0, \phi(w_i, w_j) = 0$, respectively. Therefore, the Xiangqi game $(n, \mathcal{W}, \mathcal{M})$ is a symmetric zero-sum functional-form game (FFG) (Balduzzi et al., 2019b), where players seek to maximize their own payoff while minimizing the payoff of their opponents.

***Nash equilibrium.*** The Nash equilibrium concept is widely used as a solution concept in symmetric zero-sum games (Nash, 1951). It describes a state where no player is incentivized to change their strategy from the equilibrium strategy unilaterally. However, multiple Nash equilibria can exist in a game. To guarantee the uniqueness of the Nash equilibrium, we can solve for it using the maximum entropy method (Ortiz et al., 2007). To obtain the unique maximum entropy Nash equilibrium, we can solve a Linear Programming problem as follows.

$$
\begin{aligned}
\mathfrak{p}^\star = \arg\max_{\mathfrak{p}} \sum_{j \in k} -\mathfrak{p}_j \log \mathfrak{p}_j, & \\
s.t. \quad \mathcal{M}\mathfrak{p} \leq \mathbf{0}_k, & \\
\mathfrak{p} \geq \mathbf{0}_k, & \\
\mathbf{1}_k^T \mathfrak{p} = 1. &
\end{aligned}
\tag{2}
$$

The symbols $\mathbf{0}_k$ and $\mathbf{1}_k$ represent vectors with $k$ entries, where all entries in $\mathbf{0}_k$ are equal to 0 and all entries in $\mathbf{1}_k$ are equal to 1.

***Nash Clustering.*** The concept of Nash clustering is an important tool for measuring non-transitivity. This method is based on the layered game geometry proposed in (Czarnecki et al., 2020). According to this geometry, a game's strategy space can be partitioned into an ordered list of layers.

**Definition 4.1 (The Layered Game Geometry (Czarnecki et al., 2020))** *Given a game, if the set of strategies $\Pi$ can be factorized into $k$ layers $L_i$ such that $\cup_i L_i = \Pi$, and for any layers $L_i, L_j \in L(i \neq j)$, $L_i \cap L_j = \emptyset$ , then layers are fully transitive and there exists $z \in \mathbb{R}$ such that for each $i < z$ we have $|L_i| \leq |L_i + 1|$ and for each $i \geq z$ we have $|L_i| \geq |L_i + 1|$.*

The phenomenon of non-transitivity typically does not occur within a single layer of the layered game geometry. Therefore, to measure non-transitivity, the Nash clustering method was proposed to identify mixed Nash equilibria across multiple layers.

**Definition 4.2 (Nash Clustering)** *Given a finite two-player zero-sum symmetric game and corresponding strategy set $\Pi$, Nash clustering $C := (N_j : j \in \mathbb{N} \bigwedge N_j \neq \emptyset)$, where for each $i \geq 1$, $N_{i+1} = supp(Nash(\mathcal{M}|\Pi \setminus \bigcup_{j \leq i} N_j))$ for $N_0 = \emptyset$.*

The size of strategies in each Nash cluster could be used to evaluate non-transitivity.

***Rock-Paper-Scissor Cycles.*** The formation of cycles among the strategies in the strategy space characterizes non-transitive games. A common and efficient method to measure non-transitivity is by calculating the length of the longest cycle in the strategy space. However, this is a known NP-hard problem in computing the longest directed paths and cycles in a directed graph (Björklund et al., 2004). As an alternative, we can measure non-transitivity by finding cycles of length three, known as the Rock-Paper-Scissors cycles.

**Theorem 4.1 (Rock-Paper-Scissor Cycles)** *Given a payoff matrix $\mathcal{M}$, we can obtain the corresponding adjacency matrix $\mathcal{A}$ from $\mathcal{M}$. For each element $\mathcal{A}_{i,j}$ in $\mathcal{A}$,*

$$\mathcal{A}_{i,j} = \left\{ \begin{array}{ll} 1, & \mathcal{M}_{i,j} > 0, \\ 0, & otherwise. \end{array} \right. \tag{3}$$

*$(\mathcal{A}^k)_{i,j}$ is the number of $k$-length paths from node $n_i$ to node $n_j$. The number of Rock-Paper-Scissor (RPS) Cycles in a strategy set can be obtained by the diagonal of $\mathcal{A}^3$.*

***Elo rating system.*** The Elo rating system (Elo, 1978) is frequently used to assess the relative skill levels of players in zero-sum games such as Chess or Xiangqi. It is based on the assumption that the performance of a game player is a random variable that follows a normal distribution, forming a bell-shaped curve over time. As a result, the mean value of players' performances remains at a constant level and changes gradually over time. The calculation details of the Elo rating system are provided in Appendix A.

## 4.2 Non-transitivity Analysis

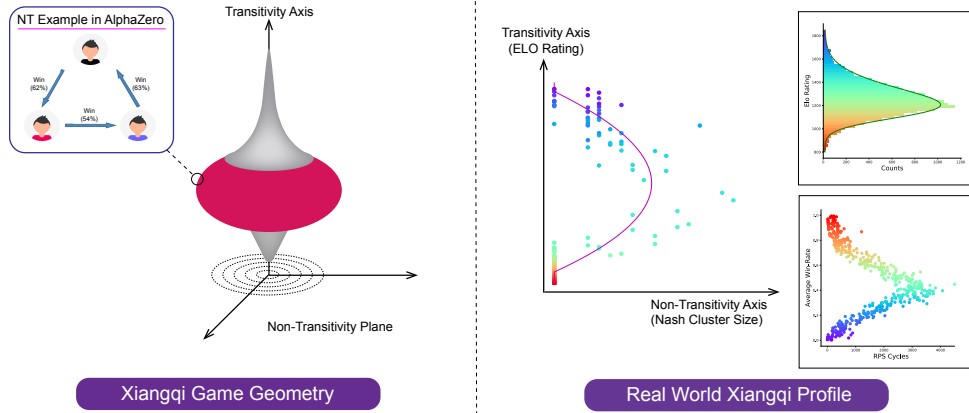

Figure 3: The Spinning Top Geometry and Profile of Xiangqi. The left figure illustrates the game geometry of Xiangqi, which resembles a spinning top structure. The upper-left corner of the left figure provides a non-transitive example of training checkpoints in AlphaZero Xiangqi. The right figures depict the real-world Xiangqi profile. In the middle sub-figure, the x-axis shows the Nash cluster size, which reflects the strength of non-transitivity, and the y-axis shows the ELO rating, which reflects the strength of transitivity. The histogram of ELO ratings (upper-right corner) and RPS cycles (lower-right corner) compares the degree of transitivity and the degree of non-transitivity, where the number of RPS cycles measures the latter.

Real-world games, including Chess and AlphaStar, are believed to have game geometrical structures that resemble a spinning top geometry, which allows the games to be split into transitive and non-transitive parts (Czarnecki et al., 2020). Interestingly, we discovered that non-transitivity also exists in the self-play of Xiangqi AlphaZero, as demonstrated in the upper-left corner of Fig. 3. The three cartoon characters, represented by different colors (black, red, and blue), correspond to the different AlphaZero agents generated sequentially by self-play. The three agents exhibit non-transitivity, where the black player wins against the red player with a probability of 0.62, and the red player wins against the blue player with a probability of 0.54. Still, the red player wins against the black player with a probability of 0.63. To calculate the win probability, we played each agent against each other at least 100 times.

To thoroughly analyze Xiangqi's geometry, we obtained a dataset consisting of over 10,000 records of real-world Xiangqi games, which were sourced from the Play Ok game platform. This online gaming platform allows users to play various games, including Chess, Xiangqi, and Checkers, with players from all over the world. Our first step in analyzing this dataset was constructing a payoff matrix by discretizing the entire strategy space, a process we outline in Algorithm 1, as illustrated in Appendix B. As the data was sourced from

human players, we used binned ELO ratings to measure the strength of transitivity. Specifically, we consider a two-way match-up to occur between any two ELO rating bins $b_i$ and $b_j$, where $b_k = [b_k^L, b_k^H]$ for $k \in i, j$, if one player's ELO rating falls into $b_i$, the other player's rating falls into $b_j$, and each player plays both black and white.

In the initial stage, we select all relevant records ($\mathcal{D}^\star$) from the dataset $\mathcal{D}$, which satisfy the Elo rating condition for players belonging to bins $b_i$ and $b_j$. In case none of the records satisfy the condition, we approximate the expected payoff score using the winning probability predicted by the Elo rating, i.e.,

$$E_{ij} = 2 \times p(i > j) - 1 = 2 \times \frac{1}{1 + \exp\left(-\frac{ln(10)}{400}\left(\frac{b_i^H - b_i^L}{2} - \frac{b_j^H - b_j^L}{2}\right)\right)} - 1. \tag{4}$$

If all the records satisfy the condition, we calculate the expected payoff score using the average game score of the games played. The game score is assigned a value of 1, 0, and -1 for a win, tie, and loss, respectively. Subsequently, we exchange players $i$ and $j$ and calculate the expected payoff score $E_{ji}$. The value of players $i$ and $j$ in the payoff matrix $\mathcal{M}$ for the two-way match-up between bin $b_i$ and bin $b_j$ is determined by averaging the expected payoff score of both cases. Specifically, we can write $\mathcal{M}i,j = \frac{E_{ij} + E_{ji}}{2}$ and $\mathcal{M}j,i = -\mathcal{M}i,j$. Therefore, the skew-symmetric payoff matrix can be constructed from real-world Xiangqi records by traversing every possible pair of bins.

The results of the analysis of the payoff matrix $\mathcal{M}$ from real-world Xiangqi records are presented in Fig. 3. The left sub-figure depicts the Xiangqi Game Geometry, which bears a resemblance to a spinning top structure. The non-transitivity plane is represented by the x-y plane, while the z-axis represents the transitivity level. Non-transitivity in the plane is increased by radiation from the origin to the outside in sequence. The right sub-figures describe the details of the real-world Xiangqi profile. The main subfigure shows the strength of non-transitivity at each ELO level, where the size of the Nash cluster size quantizes the non-transitivity. The blue curve, which illustrates the non-transitivity game profile, is obtained by fitting a skewed-normal curve to the Nash cluster size. It verifies the spinning top game geometry hypothesis. Similar to the 3D game geometry, the middle region of the ELO rating in the transitivity axis is accompanied by much stronger non-transitivity. Furthermore, the strength of non-transitivity (Nash cluster size) gradually diminishes with an increase or decrease in ELO rating. Similar evidence is revealed in the right sub-figures of the Real World Xiangqi Profile. These subfigures show the comparison between the histogram of transitivity strength (right-up corner) and the non-transitivity strength (right-bottom corner) at each average win rate, which is measured by ELO rating and the number of Rock-Paper-Scissor cycles. The middle region between 0.4 and 0.6 indicates a higher degree of non-transitivity, which is consistent with the spinning top game geometry and real-world Xiangqi profile presented in Fig. 3. As the average win rate increases or decreases, the strength of non-transitivity (the number of Rock-Paper-Scissor cycles) gradually diminishes.

## 5 JiangJun: the Method

### 5.1 JiangJun Algorithm

Through the analysis of human Xiangqi gameplay data, we have discovered that Xiangqi exhibits a spinning top game geometry structure, which reveals the presence of non-transitivity. To address this non-transitivity issue, we propose the JiangJun algorithm. As depicted in Figure 4, JiangJun is comprised of two primary modules, the MCTS Actor and the Populationer. The MCTS Actor module generates training data, while the Populationer module maintains a Population consisting of varied policies, an inference module, and a maximum entropy Nash solver.

***Populationer.*** The proposed *Populationer* module maintains a population of JiangJun policies and selects opponents that are approximate best responses to the Nash mixture of the population. The algorithmic details of *Populationer* are presented in Algorithm 2, as illustrated in Appendix B. Specifically, *Populationer* keeps track of two entities: a population of previous JiangJun policies, and a Nash buffer for storing game results between the latest best response $\mathfrak{J}$ and all agents in the population. The information of the game result is stored as a tuple $(n_0, n_1, n_2)$, where $n_0$ denotes the game score for the red player, which is assigned as $1, 0, -1$ for the wins, ties, and losses of the red, respectively. $n_1$ and $n_2$ represent the names of the red and

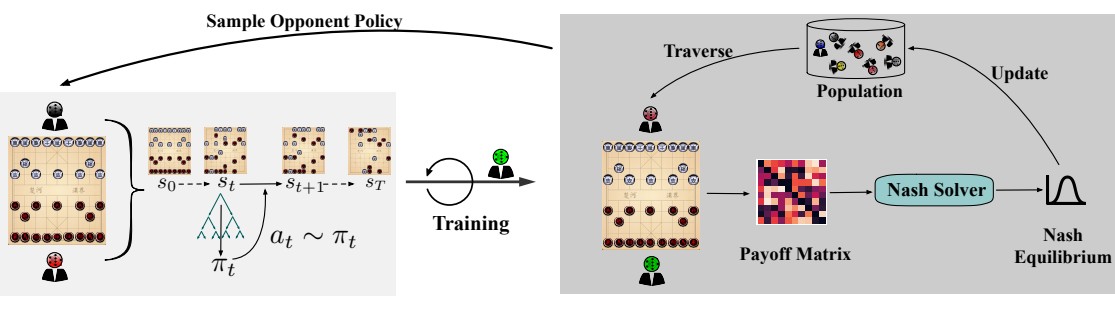

A. MCTS Actor                                    B. Populationer

Figure 4: The proposed JiangJun algorithm consists of two main modules: *MCTS Actor* and *Populationer*. The *MCTS Actor* is the inference module that generates trajectories used for training by sampling policies from the population according to the Nash equilibrium distribution. The resulting best response policies are then fed back to the *Populationer*. In the *Populationer*, new policies are tested against each policy in the Population, and the payoff matrix is completed. We use a maximum entropy Nash solver to derive the Nash equilibrium from the payoff matrix, which is then used for Population update and policy selection.

black players. During inference, the red and black players alternate positions in each game until $k$ games are played.

Once the Nash buffer is filled, the payoff matrix $\mathcal{M}$ is constructed by loading every tuple of the game's result into the buffer. For each tuple $(n_0, n_1, n_2)$, we add $n_0$ to the value of $\mathcal{M}_{n_1,n_2}$ and subtract $n_0$ from the value of the symmetric position $\mathcal{M}_{n_2,n_1}$. Thus, the payoff matrix is anti-symmetric. We obtain the maximum entropy Nash equilibrium of $\mathcal{M}$ by using the maximum entropy Nash Solver and Equation 2. The resulting equilibrium distribution is used to select an appropriate opponent from the top-$n$ agents in the population with a higher probability. Additionally, the JiangJun agent with the lowest probability is eliminated from the population to manage computing resources and replaced by the latest JiangJun agent $\mathfrak{J}$. The proposed *Populationer* module is illustrated in the right subfigure of Fig. 10.

**MCTS Actor.**     The *MCTS Actor* module plays a crucial role in our proposed method. Once an opponent has been selected, the latest updated agent $\mathfrak{J}$ and the opponent agent are provided as inputs to the module, as illustrated in the left subfigure of Fig. 4. The latest JiangJun agent uses the Monte Carlo Tree Search (MCTS) algorithm to play against the opponent agent for the current state $s$. The position information of each piece is represented using a $0, 19 \times 10 \times 14$ matrix. The schematic diagram of state $s$ can be found in Appendix C. Starting from the root node $s$, *MCTS Actor* begins a sequence of simulations. At each turn $t$, the *MCTS Actor* selects an action $a_t$ in the current state $s_t$ until the leaf node is reached. Here, a ResNets-based neural network within the JiangJun agent predicts the action probability $\mathbf{p}$ and value $v$ with parameters $\theta$ as follows: $(\boldsymbol{p}, v) = f\theta(s)$, where $f_\theta$ is the JiangJun network, $s$ represents the state, $\boldsymbol{p} = [p_{a_1}, p_{a_2}, \cdots]$ denotes a vector of action probabilities, and $v$ approximates the expected outcome $z$ of the Xiangqi game from state $s$. Each prior probability $p_a$ is assigned according to the corresponding predicted action probability, and the leaf node is expanded.

After a series of simulations from the root node $s$, the *MCTS Actor* generates the actions probability distribution $\boldsymbol{\pi}$, where each component represents the probability of each action. Subsequently, a move is chosen according to $a \sim \boldsymbol{\pi}$, and the game proceeds until its conclusion. Finally, the final score $z$ is determined, with $+1$ indicating a win, $0$ indicating a tie, and $-1$ indicating a loss. Consequently, each trajectory contains information from every turn, including the state $s$, the search probabilities $\boldsymbol{\pi}$, the expected result $z$, and the predicted value $v$ and the predicted action probabilities $\boldsymbol{p}$. Each trajectory is saved in the Replay Buffer after the game.

**Training.**     Within our method, we employ a *Training* module to update the parameters of the JiangJun agent by simultaneously sampling the trajectories of the replay buffer and participating in a training process between *MCTS Actor* and *Populationer*. To achieve this, we employ a mean-square loss function that minimizes the error between the predicted value $v$ and the actual outcome $z$ of the game and a cross-entropy

loss function that maximizes the similarity between the probabilities of predicted actions $\boldsymbol{p}$ and the search probabilities $\boldsymbol{\pi}$. This process can be expressed mathematically as follows.

$$l = (z - v)^2 - \alpha \boldsymbol{\pi}^T \log \boldsymbol{p} + \beta \|\theta\|^2, \tag{5}$$

where $\alpha, \beta$ are balance constants between 0 and 1, and $\|\theta\|^2$ is the $L_2$ weight regularization of JiangJun agent.

## 5.2 Computational Analysis

This section is dedicated to a comprehensive dissection of the computational complexity associated with the two key components of Jiangjun: the *MCTS Actor* and the *Nash Solver*.

Beginning with the MCTS Actor, it manifests a worst-case time complexity that can be expressed as $O(b^d \times n \times C)$. In this expression, $b$ symbolizes the effective branching factor, $d$ corresponds to the effective search depth, $n$ represents the number of simulations, and $C$ is indicative of the neural network's inference time. In the context of Xiangqi, the median branching factor fluctuates between 30 and 80, contingent upon the specific position in question. Furthermore, the game tree depth for Xiangqi has the potential to reach beyond 40. Conventionally, the number of simulations $n$ is determined to be 800 or 1800. The neural network's inference time $C$ is subject to modifications by factors including but not limited to network size, architecture, implementation efficiency, and the utilization of hardware accelerators such as GPUs or TPUs. It is of paramount importance to highlight that the time complexity $O(b^d \times n \times C)$ should be interpreted as a worst-case approximation, whereas in practical scenarios, the time complexity often falls substantially below this estimate.

Shifting focus to the Nash Solver, it calculates a unique maximum entropy Nash equilibrium by addressing a Linear Programming (LP) problem, as elucidated in Eq. 2 in our manuscript. During the $n$th iteration, both player 1 and player 2 have $k$ strategies at their disposal. The time complexity of resolving an LP problem is dependent on the specific algorithm invoked. For instance, the Simplex method exhibits a worst-case time complexity of $O(2^k \times poly(k))$, wherein $poly(k)$ denotes a polynomial function of $k$.

## 5.3 Evaluation Metrics

In a non-transitive game, the performance improvement of one player may not be informative, as other players can still beat that player. To address this issue, we propose a relative population ELO rating (RP-ELO) measure to evaluate the JiangJun agent's progress. At the beginning of the study, the initial ELO rating of all agents in the population and the updated JiangJun agent $\mathfrak{J}$ then is set to 1500. The JiangJun agent $\mathfrak{J}$ plays 100 games with each agent in the population to update their ELO ratings. As such, the RP-ELO provides a more comprehensive reflection of the population's improvement, rather than relying solely on the last agent, as it is non-transitive, but the ELO rating increases.

In addition, we employ the quantitative metric Relative Population Performance (RPP) (Balduzzi et al., 2019b) to assess the performance of the population and the degree to which JiangJun successfully addressed the non-transitivity issue.

**Definition 5.1** *Consider two populations $\mathcal{A}$ and $\mathcal{B}$, where each population comprises a set of agents. Let $(p, q)$ be the Nash equilibrium for a zero-sum game on the payoff matrix $\mathcal{M}_{\mathcal{A},\mathcal{B}}$. The relative population performance is*

$$\mathbf{v}(\mathcal{A}, \mathcal{B}) := p^T \cdot \mathcal{M}_{\mathcal{A},\mathcal{B}} \cdot q. \tag{6}$$

*Suppose $\mathbf{v}(\mathcal{A}, \mathcal{B})$ is greater than 0. In that case, it suggests that population $\mathcal{A}$ has achieved a significant performance improvement in the absence of non-transitivity when compared to population $\mathcal{B}$.*

Exploitability is a measure used to evaluate how closely a strategy profile approximates a Nash equilibrium (Timbers et al., 2022). A lower exploitability value signifies a strategy that is closer to being optimal. In formal terms, given suboptimal strategies $\pi$ for $n$ players, exploitability can be calculated using the following formula:

$$\text{Exploitability}(\pi) = \frac{1}{n} \sum_i \left( u_i(br(\pi_{-i}), \pi_{-i}) - u_i(\pi) \right), \tag{7}$$

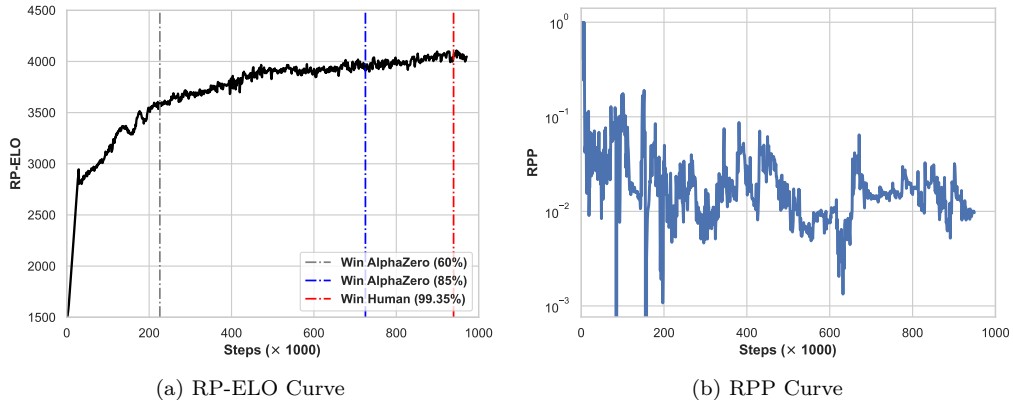

(a) RP-ELO Curve          (b) RPP Curve

Figure 5: The figure illustrates the training progress of the JiangJun algorithm, with the Relative Population ELO Rating (RP-ELO) and Relative Population Performance (RPP) plotted in the left and right subfigures, respectively. The vertical dashed lines on the RP-ELO curve indicate the specific stages when the JiangJun agent competes with AlphaZero and human players with different win rates. The RPP curve shows the performance improvement of the population as a function of training steps (×1000). A positive RPP value indicates that the population has meaningfully improved without non-transitivity.

where $u$ represents the reward function and $br$ denotes the best response.

In the context of zero-sum games, the second terms in the equation sum to zero. As a result, the equation can be simplified to:

$$\text{Exploitability}(\pi) = \frac{1}{n} \sum_i u_i(br(\pi_{-i}), \pi_{-i}). \tag{8}$$

However, calculating the exploitability of large-scale games is nearly infeasible. In this study, we employ the standard AlphaZero algorithm to approximate the exact best response.

## 6 Experiment Results

The training of the JiangJun algorithm was performed on the Huawei Cloud ModelArt platform, utilizing a total of 90 V100 GPUs. Specifically, 78 of these GPUs were allocated for the *MCTS Actor*, 4 GPUs were used for the *Training*, and 8 GPUs were dedicated to the *Populationer*. A detailed description of the training framework can be found in Appendix D. To comprehensively evaluate JiangJun, we developed an interactive WeChat mini program that allows human players to compete against the agent in Xiangqi games. WeChat mini-programs are sub-applications that operate within the WeChat ecosystem, which currently boasts over a million, daily active users. Appendix E provides more details on the JiangJun mini program. In Section 6.1, we present experimental results demonstrating that JiangJun can overcome the non-transitivity problem. Furthermore, in Section 6.2, we report on the agent's performance against human players in the mini-program. A case study of endgame playing is given in Section 6.3.

### 6.1 Experiment Results: Conquering the Non-transitivity

**RP-ELO and Win Rate.** The training progress of the JiangJun algorithm is presented in Fig.5, where we utilize the Relative Population ELO Rating (RP-ELO) and Relative Population Performance (RPP) to evaluate its performance. The left figure in Fig.5 shows the increase of RP-ELO with the training steps (×1000). We also indicate three vertical dashed lines from left to right as the milestones of JiangJun strength with the standard AlphaZero Xiangqi algorithm, which has achieved a strength comparable to a 9-dan player. The first dashed line indicates that the JiangJun agent at 227k steps could win Xiangqi AlphaZero with a 60% win probability. The win probability increases to 85% at 726k steps, shown as the middle vertical

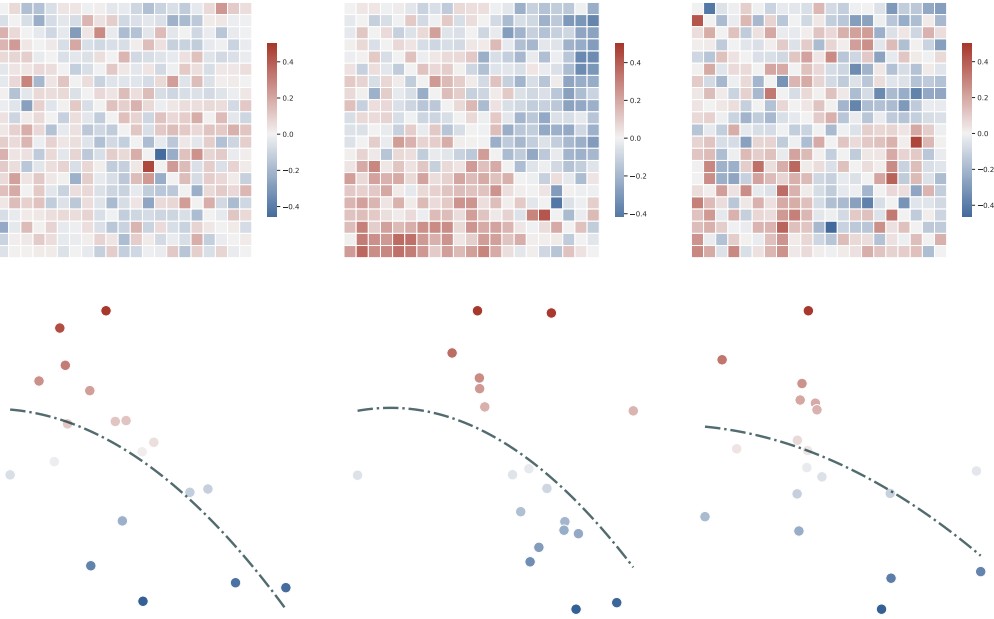

Figure 6: This figure displays the Low Dimension Gamescapes of JiangJun, with the top row of each column showing the payoff matrix of populations consisting of 21 agents. The color variation, ranging from blue (-0.5) to red (0.5), corresponds to the average game score of an agent against each other. The bottom row of each column presents the points corresponding to the first two-dimensional embedding of Shur decomposition. Additionally, a dash-dotted line is plotted for the second-order linear regression, with its outliers detected and deleted by Z-Score.

dashed line. At 940k steps, JiangJun agent can win 99.35% human players in our JiangJun mini-program. The win probability with human players is calculated based on nearly 1300 games.

On the other hand, we offer an additional strength evaluation by comparing our approach with the behavior cloning Xiangqi algorithm, shorted as BC Xiangqi. Our JiangJun model has demonstrated the ability to defeat BC Xiangqi with a win rate of 96.40%, as determined from a sample of more than 100 games. The specifics of the behavior cloning model for Xiangqi are outlined below. To train the model, we collected and processed a dataset consisting of 300,000 Xiangqi data samples. Each sample is composed of an input-output pair $(s, a)$, where input $s$ represents the state as a $9 \times 10 \times 14$ binary matrix, and output $a$ is a one-hot action vector with 2048 dimensions. The state representation is identical to that used in our JiangJun model. Regarding the state representation, each of the 14 planes is a $9 \times 10$ matrix, with the first seven planes representing the positions of the red player's pieces and the last seven planes representing the positions of the black player's pieces. We utilized the ResNet-18 architecture (He et al., 2016) as the basis for our behavior cloning approach. The ResNet-18 model takes the state as input and predicts the corresponding action.

**RPP.** In addition, we scrutinize the Relative Population Performance (RPP) as illustrated in the right figure of Fig. 5. Initially, the RPP value witnesses a sharp ascension, indicative of significant performance improvement at the onset of this phase. Following this rapid rise, the RPP value achieves a state of equilibrium, maintaining a level consistently above zero, with the exception of data points at 13k and 14k training steps. This fluctuation, though minor, is an essential detail to acknowledge as it provides insights into the evolving performance metrics during training. The trajectory of the RPP value curve essentially underscores the efficacy of our proposed JiangJun methodology. In the face of non-transitivity, a common obstacle in such domains, JiangJun demonstrates robustness, ensuring the consistency of population performance improvement. This analysis not only validates the effectiveness of our method in enhancing performance but also illustrates the dynamic nature of learning progression. JiangJun's capability to maintain consistent improvement,

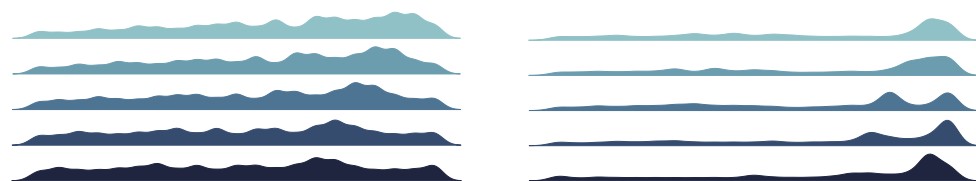

Figure 7: Comparison of Nash equilibrium distributions between AlphaZero (on the left) and JiangJun (on the right). Each row of subfigures represents the Nash distribution of a single population at approximately 115k training steps, with the agent index increasing from left to right.

despite encountering the complexities of non-transitivity, speaks volumes about its practical applicability and efficiency.

**Exploitability.** We additionally measure the exploitability of our JiangJun algorithm and AlphaZero Xiangqi algorithm. This exploitability metric is a measure of an algorithm's susceptibility to being exploited by an adversary, where a lower value signifies a strategy closer to optimality. Our proposed JiangJun method displays a marked improvement in performance, demonstrating an exploitability value of merely 8.41%, significantly lower than the 25.53% of AlphaZero Xiangqi. The execution of these experiments relied on the power of high-performance computing, specifically utilizing 40 V100 GPUs. The approximate best responses used for the computation of exploitability were trained for about 150,000 steps. These results clearly highlight the superior performance of JiangJun in minimizing exploitability, demonstrating its effectiveness compared to conventional strategies, particularly when powerful computational resources are employed.

**Visualization Results.** This section presents two case studies of the JiangJun training process, demonstrating its effectiveness in overcoming non-transitivity in Xiangqi. The first case study presents the visualization results of our approach to conquering non-transitivity, as shown in Figure 6. The figure displays three sets of low-dimensional gamescapes as the training progresses from left to right. The top row of each set represents the payoff matrix of populations consisting of 21 agents. The color gradient from blue to red represents the average game score of the corresponding agent against other agents, where a bluer color indicates a lower win rate (i.e., a score closer to -0.5), and a redder color indicates a higher win rate (i.e., a score closer to 0.5). The bottom row of each set shows the corresponding 2D embedding of the Shur decomposition, and the dashed-dotted line denotes the second-order linear regression. Outliers in the linear regression are identified and removed using the Z-score technique.

In an ideally transitive game, the gamescape will appear as a line in the 2D embedding figure, while a non-transitive game will form a cycle (Balduzzi et al., 2019b). As illustrated in Figure 6, the second order linear regression is almost a straight line across the three different training steps, suggesting that the JiangJun agent's training is approaching transitivity. This finding indicates that our approach effectively conquers non-transitivity, which ensures consistent performance improvement across the population.

The probability distributions of the Nash equilibria of AlphaZero and JiangJun are illustrated in Figure 7. Each row of subfigures represents the Nash distribution of a single population at approximately 115k training steps, with the agent index increasing from left to right. The left subfigure in Fig.7 shows that the Nash probabilities are nearly uniformly distributed when AlphaZero is trained for about 115k steps. In contrast, the right subfigure of Fig.7 indicates that the Nash probabilities of the latest agent updated by the JiangJun algorithm dominate the entire population of agents.

## 6.2 Experiment Results: Human-AI Experiments

In order to evaluate the effectiveness of our JiangJun algorithm in a comprehensive manner, we established a JiangJun mini-program on the WeChat platform, with further specifics available in Appendix E. Over the course of six months of deployment, the mini-program allowed us to accumulate 7061 game records against human players, as documented in Table 2. These statistics, recorded on a monthly basis, enumerate JiangJun's victories, draws, losses, total matches, and the win rates corresponding to each month.

| Deployment Time | Stage | Wins | Ties | Losses | Total | Win Rate |
|:---:|:---:|:---:|:---:|:---:|:---:|:---:|
| Month 1 | Training | 717 | 11 | 8 | 736 | 97.42% |
| Month 2 | Training | 724 | 0 | 17 | 741 | 97.71% |
| Month 3 | Training | 462 | 0 | 3 | 465 | 99.35% |
| Month 4-6 | Evaluation | 5089 | 3 | 27 | 5119 | 99.41% |

Table 2: Monthly statistics of the JiangJun mini-program over a six-month period are presented in this table. The data is divided into two stages: Training and Evaluation. The Training stage denotes the period during which the deployed weights undergo training, while the Evaluation stage uses well-trained weights. The table provides information on the number of games won, tied, and lost by JiangJun against human players, as well as the total number of games played and the win rate for each month. It should be noted that the initial month listed corresponds to the commencement of the mini-program deployment, rather than the onset of the training phase.

The table also distinguishes the performance data based on two key deployment stages of the JiangJun algorithm, referred to as "Training" and "Evaluation". The "Training" stage represents a period during which the weights of the algorithm underwent training, whereas the "Evaluation" stage leverages the optimally trained weights. It should be noted that the initial month listed in Table 2 corresponds to the commencement of the mini-program deployment, rather than the onset of the training phase. Throughout the "Training" phase, which covered the initial three months, the JiangJun agent demonstrated a laudable performance, averaging a win rate of 98.16%, thus indicating its considerable prowess even during its learning and adaptation phase. Upon transitioning to the "Evaluation" stage in the subsequent three months, the mini-program recorded an average of approximately 1703 games each month, while JiangJun sustained a remarkable win rate, averaging 99.41%. These observations underscore the JiangJun agent's extraordinary capability in effectively engaging with human players in the strategic game of Xiangqi.

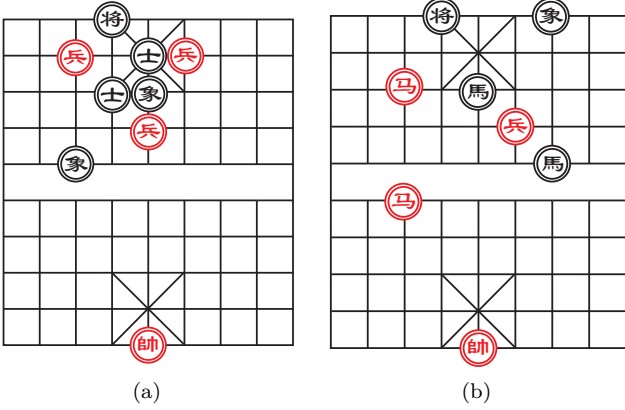

(a)       (b)

Figure 8: Two examples of the endgame: "Three Pawns V.S. the full Advisors and Bishops" in the left figure, "double Knight and Pawn V.S. double Knight and single Bishop" in the right figure. Details are provided in Appendix F.

## 6.3 Experiment Results: Case Study of Endgame Playing

The endgame in Xiangqi is a crucial phase that marks the culmination of the game, wherein both sides engage in a final struggle for survival. Fig. 8 illustrates the initial states of two Xiangqi endgame scenarios. As this stage features fewer pieces on the board and less variation in tactics, it becomes increasingly difficult for AI models to master fundamental yet practical skills needed to solve the endgame. Despite these challenges, our analysis of the JiangJun algorithm's trajectories reveals that it has succeeded in learning how to solve the Xiangqi endgame effectively.

For instance, the classical endgame of "three Pawns V.S. the full Advisors and Bishops" is one such example where JiangJun has demonstrated its prowess in winning the endgame by keeping one Pawn in a higher position and using the other two to form a left and right pincer attack. The trajectories of JiangJun's successful endgame strategies are displayed in Appendix F, highlighting the trajectories of the "double Knight and Pawn V.S. double Knight and single Bishop" endgame. Our analysis of JiangJun's trajectories demonstrates that it has successfully learned the key strategies to win Xiangqi endgames.

## 7    Conclusion

This study delves into the intricate geometry of Xiangqi, employing a substantial dataset of over 10,000 human gameplays to unearth significant non-transitivity in the middle region of transitivity, reflected in a spinning top-like structure. Our research further highlights the presence of cyclical strategies within AlphaZero's Xiangqi training checkpoints. Addressing this non-transitivity issue, we introduce the JiangJun algorithm, an innovative approach diverging from the self-play tactic of AlphaZero, utilizing Nash response for opponent selection during training. Empirically, JiangJun achieves a Master level in Xiangqi, with an exceptional 99.41% win rate against human opponents on our developed WeChat mini-program and proficient handling of complex Xiangqi endgames. Moreover, assessments of relative population metrics, exploitability, and visual representations reinforce JiangJun's remarkable capability to conquer the challenges posed by non-transitivity in Xiangqi. In summary, our study offers valuable insights into Xiangqi's strategic landscape, underscoring the potency of JiangJun in navigating the game's non-transitive complexities.

## Acknowledge

Yang Li is supported by the China Scholarship Council (CSC) Scholarship. We would like to express our gratitude to Binyi Shen for his invaluable contributions to the collection of the BC Xiangqi dataset and its implementation. Our appreciation also extends to Yunkun Xu for his insightful discussions.

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

## A  Elo Rating System

Denote the players of Xiangqi as $w$ and $b$. If $w$ and $b$ has a rating of $R^w$ and $R^b$ respectively, then the expected scores are given by:

$$
\begin{aligned}
E_w &= \frac{1}{1 + 10^{(R_b - R_w)/400}}, \\
E_b &= \frac{1}{1 + 10^{(R_w - R_b)/400}}.
\end{aligned}
\tag{9}
$$

The expected scores also could be expressed as

$$
\begin{aligned}
E_{\mathrm{w}} &= \frac{Q_{\mathrm{w}}}{Q_{\mathrm{w}} + Q_{\mathrm{b}}}, \\
E_{\mathrm{b}} &= \frac{Q_{\mathrm{b}}}{Q_{\mathrm{w}} + Q_{\mathrm{b}}},
\end{aligned}
\tag{10}
$$

where $Q_{\mathrm{w}} = 10^{R_{\mathrm{w}}/400}$, and $Q_{\mathrm{b}} = 10^{R_{\mathrm{b}}/400}$. Besides, $K$-factor is another important variable for Elo rating system. The value of $K$ represents the maximum possible adjustment per game. Therefore, rating update formula of player $w, b$ are

$$
\begin{aligned}
R_w &= R_w + K\left(S_w - E_w\right), \\
R_b &= R_b + K\left(S_b - E_b\right),
\end{aligned}
\tag{11}
$$

where $S_{w/b}$ is the actually scored points. After obtaining the ELO rating, we could approximate the winning probability $p$ by

$$
p(w, b) \approx \frac{1}{1 + \exp\left(-\frac{ln(10)}{400}(R_w - R_b)\right)}.
\tag{12}
$$

## B  Algorithms

---

**Algorithm 1:** Algorithm for building the payoff matrix $\mathcal{M}$

---

**Data:** Dataset: $\mathcal{D}$, $m$ bins $B = \{b_1, b_2, \cdots, b_m\}$, $\mathcal{M} = \mathbf{0}_{m \times m}$.
**Result:** Payoff Matrix: $\mathcal{M}$.
initialization;
**for** $i \in [1, 2, \cdots, m]$ **do**
    **for** $j \in [i + 1, \cdots, m]$ **do**
        **if** $\mathcal{D}^\star = \emptyset$ **then**
            $R_i, R_j = \frac{b_i^H - b_i^L}{2}, \frac{b_j^H - b_j^L}{2}$ ;
            $E_{ij} = 2 \times p(i, j) - 1.$ ;
        **else**
            $E_{ij} = 0, n = 0$;
            **for** *data in* $\mathcal{D}^\star$ **do**
                score $= 1$ if $i$ wins, 0 if ties, -1 if $i$ losses;
                $E_{ij} = E_{ij} +$ score;
                $n = n + 1$;
            **end**
            $E_{ij} = E_{ij}/n$;
        **end**
        Exchange players $i, j$, repeat the above steps and obtain $E_{ji}$;
        $\mathcal{M}_{i,j} = \frac{E_{ij} + E_{ji}}{2}$;
        $\mathcal{M}_{j,i} = -\mathcal{M}_{i,j}$;
    **end**
**end**

---

---

**Algorithm 2:** Algorithm for *Populationer*

---

**Input:** Latest agent $\mathcal{J}$ updated by *Train*. Population $\mathfrak{A}$ with $k$ agents. Nash replay buffer $\mathfrak{B}$.

**for** *agent* $\mathfrak{a}$ *in* $\mathfrak{A}$ **do**

    // Play with every agent in *Play*

    **for** $i = 1, 2, \ldots, k$ **do**

        **if** $i\%2 == 0$ **then**

            red $\longleftarrow \mathcal{J}$.

            black $\longleftarrow \mathfrak{a}$.

        **else**

            red $\longleftarrow \mathfrak{a}$.

            black $\longleftarrow \mathcal{J}$.

        **end**

        // *Play* returns game result information

        $t =$(score $n_0$ for red (win=1,tie=0,loss=-1), red index $n_1$, black index $n_2$) $\longleftarrow$ red plays with black.

        Store game information tuple $t$ in $\mathfrak{B}$.

    **end**

**end**

Initialize payoff matrix $\mathcal{M} = \mathbf{0}_{k \times k}$.

**for** *tuple* $(n_0, n_1, n_2)$ *in* $\mathfrak{B}$ **do**

    $\mathcal{M}_{n_1,n_2} = \mathcal{M}_{n_1,n_2} + n_0$.

    $\mathcal{M}_{n_2,n_1} = \mathcal{M}_{n_2,n_1} - n_0$.

**end**

$\mathbf{p} \longleftarrow$ Nash on $\mathcal{M}$.

$P_o \longleftarrow$ Sample from top-n agents with higher probability.

$\mathfrak{A} \longleftarrow \mathfrak{A} \cup \{\mathcal{J}\} \backslash \{$agent with lowest probability.$\}$

**Output:** Opponent agent $P_o$, New population $\mathfrak{A}$.

---

## C   Representation of Game State

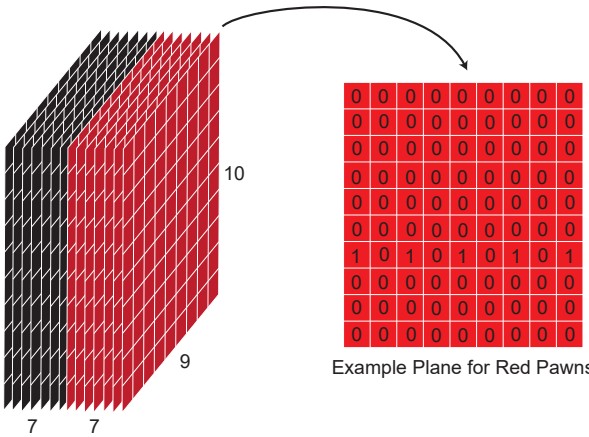

Figure 9: Schematic diagram of the game state.

The schematic diagram of state $s$ is given in Fig 9. The composition of state $s$. $s$ is a $9 \times 10 \times 14$ matrix, where each plane presents one type of piece's position. with each component of 0 (represents absence) and 1 (represents presence), as shown in the left sub-figure. In the figure, a plane with red or black color represents the piece position of red player or black player respectively. The right sub-figure shows an example of red Pawns, where the Pawn locates in a position of 1. Taking red Pawns as an example, the right figure of Figure 9 represents the initial state of red Pawns. Besides, the first 7 planes are assigned for red player and the last 7 planes are for black player.

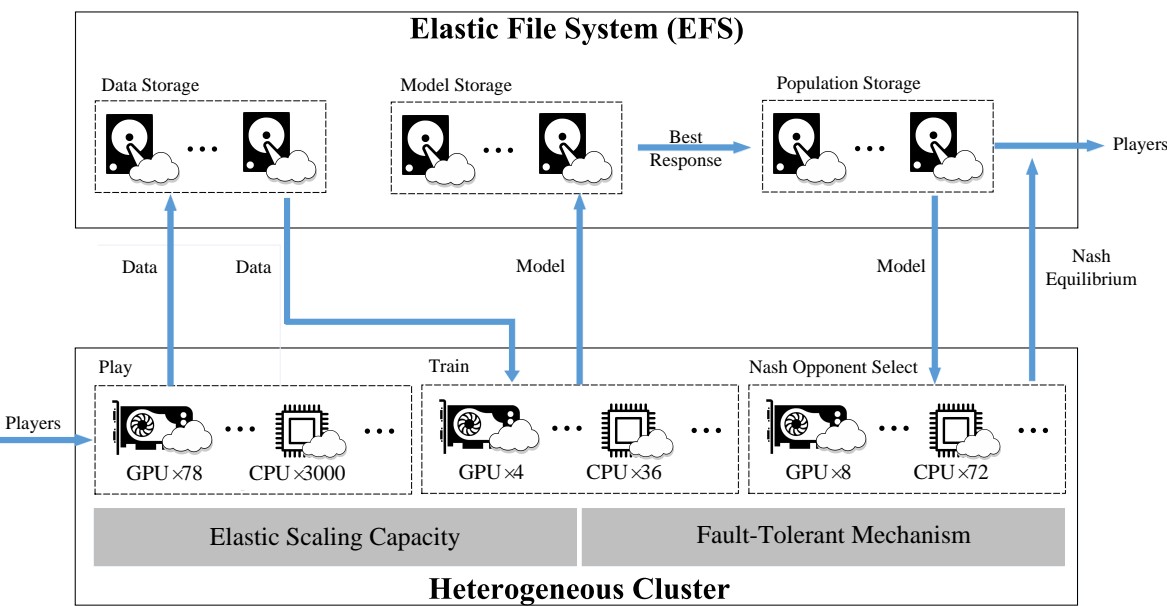

Figure 10: JiangJun Cloud-Native Training Framework.

# D  JiangJun: Training Framework and Details

## D.1  Training framework

We firstly introduce the training framework, which is conducted on the basis of Huawei Cloud ModelArt. Figure 10 gives a briefly introduction of our could-native JiangJun training framework, consisting of two main modules, i.e., heterogenous cluster (HC) and elastic file system (EFS). HC module aims to provide GPUs, CPUs heterogeneous distributed computing services. In our whole resource pool, we have 90 32GB NVIDIA V100 GPUs and over 3000 CPU cores. How to make so many resources run efficiently, safely and stably is one of the important issues of the whole HC design. Consequently, Elastic Scaling Capacity and Fault-Tolerant Mechanism are provided to ensure the efficient, secure and stable run of GPUs and CPUs.

## D.2  Training details

The hyperparameters of the network and training are provided as follows.

- network filters: 192,
- network layers: 10,
- batch size: 2048,
- sample games: 500,
- c_puct : 1.5,
- saver step: 400,
- learning rate : [0.03, 0.01, 0.003, 0.001, 0.0003, 0.0001, 0.0003, 0.001, 0.003, 0.01],
- minimum games in one block: 5000,
- maximum training blocks: 100,
- minimum training blocks: 3,
- number of the process: 10.

# E   JiangJun: Mini Program

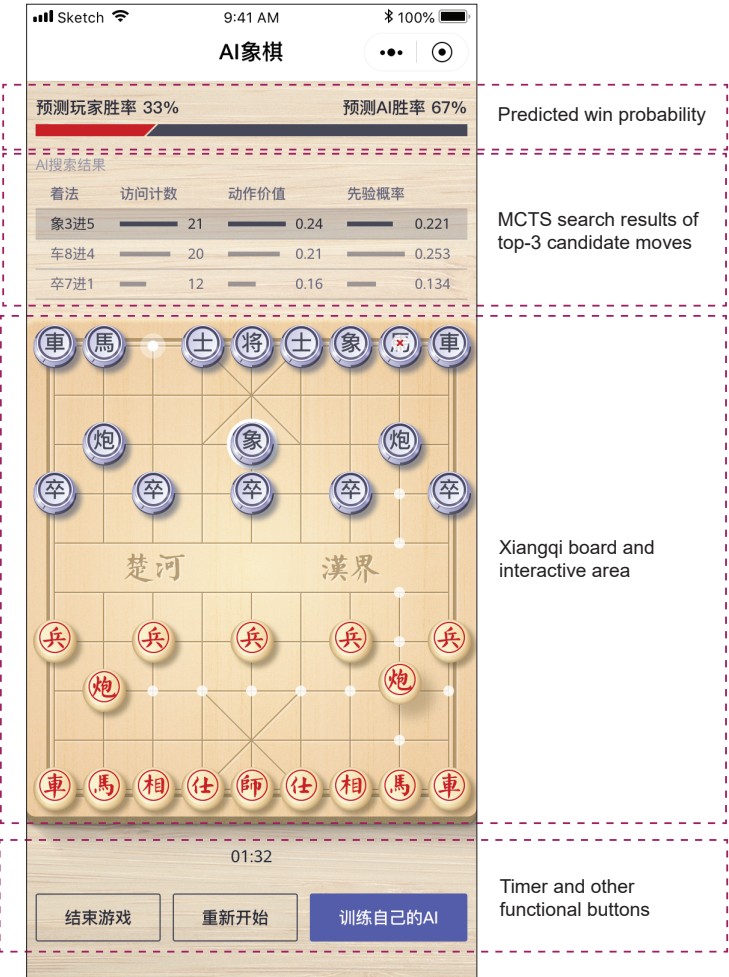

Figure 11: The JiangJun WeChat mini-program is composed of four distinct regions, including the win probability region, MCTS search information region, Xiangqi board region, and functional button region, which are all depicted in the provided screenshots.

The JiangJun mini program, depicted in Figure 11, allows users to play Xiangqi against our trained JiangJun agent. The top of the program displays the predicted win probability. The next section presents the top three candidate actions with their corresponding prior probabilities obtained from the Monte Carlo Tree Search (MCTS). For each action, we show its name in Chinese and key items used in action selection, including the visit count of the parent node $N(s)$, the action value $Q(s,a)$, and the prior probability $P(s,a)$ of selecting action $a$ in state $s_t$. The main area of the mini program is where the human player interacts with JiangJun. Finally, the bottom region contains a game timer and three function buttons: finish, restart, and train your own JiangJun AI in ModelArt.

# F  Trajectories of JiangJun Endgames Playing

## F.1  Game A : Three Pawns V.S. the full Advisors and Bishops

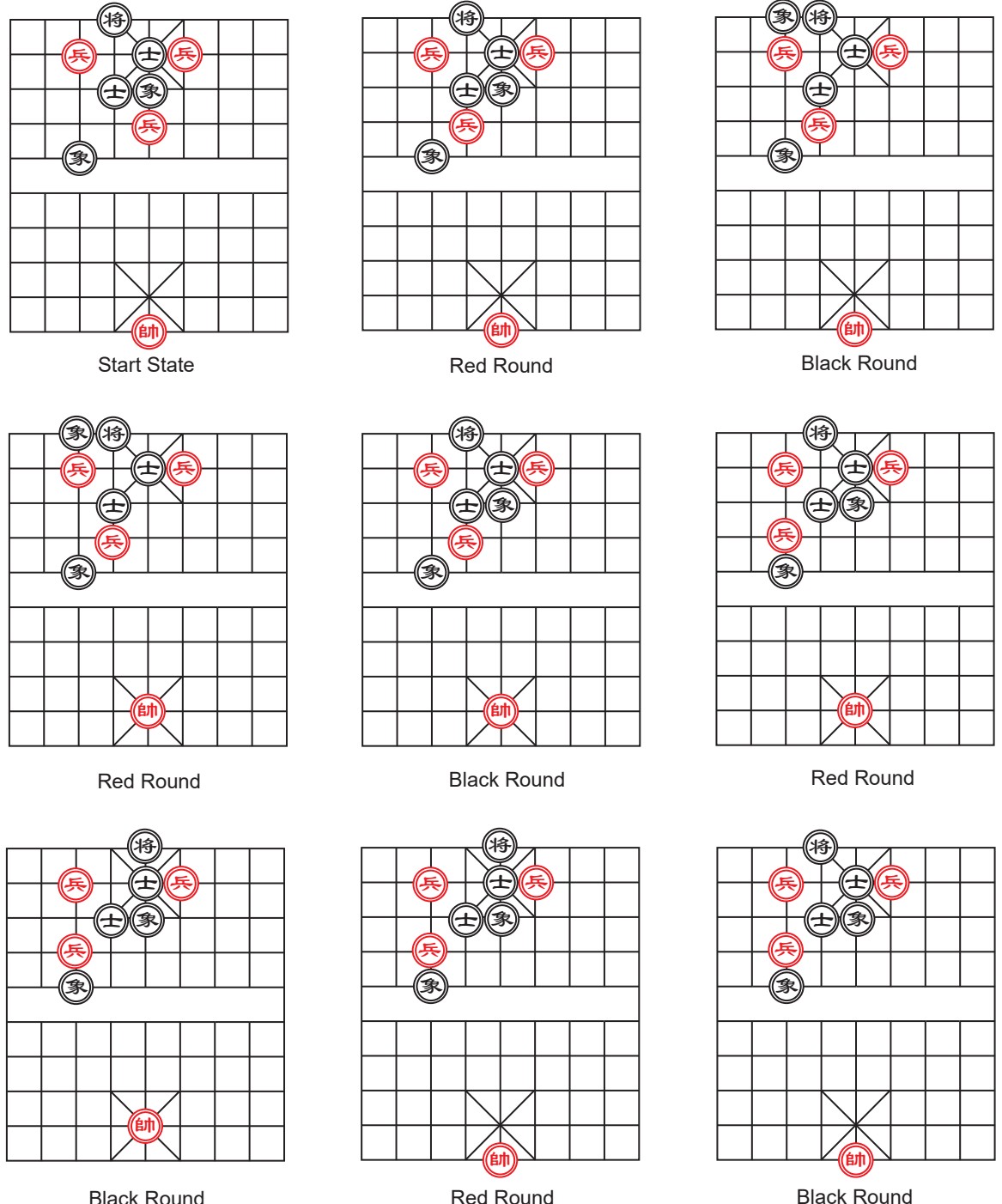

Figure 12: Game Trajectories of JiangJun playing endgame "Three Pawns V.S. the full Advisors and Bishops".

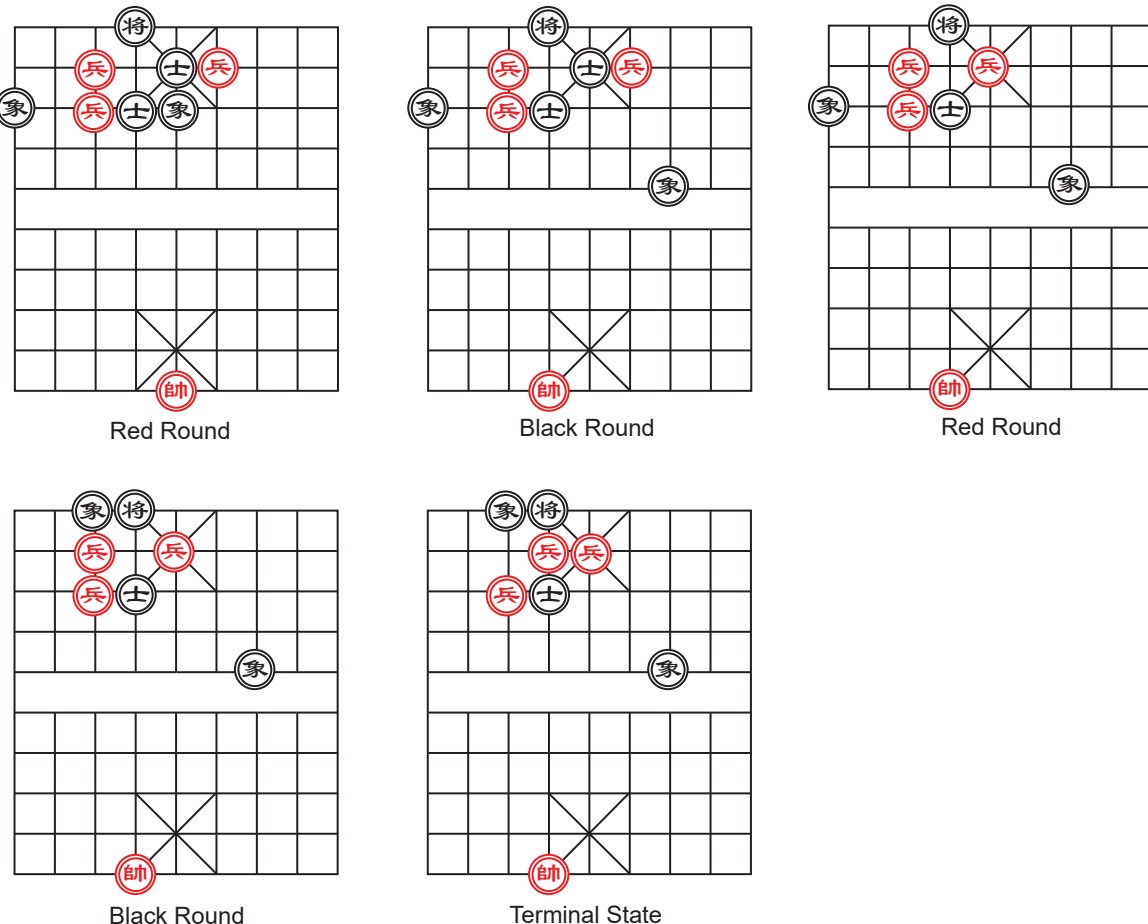

Figure 13: Game Trajectories of JiangJun playing endgame "Three Pawns and the full Advisors and Bishops".

## F.2 Game B: double Knight and Pawn V.S. double Knight and single Bishop

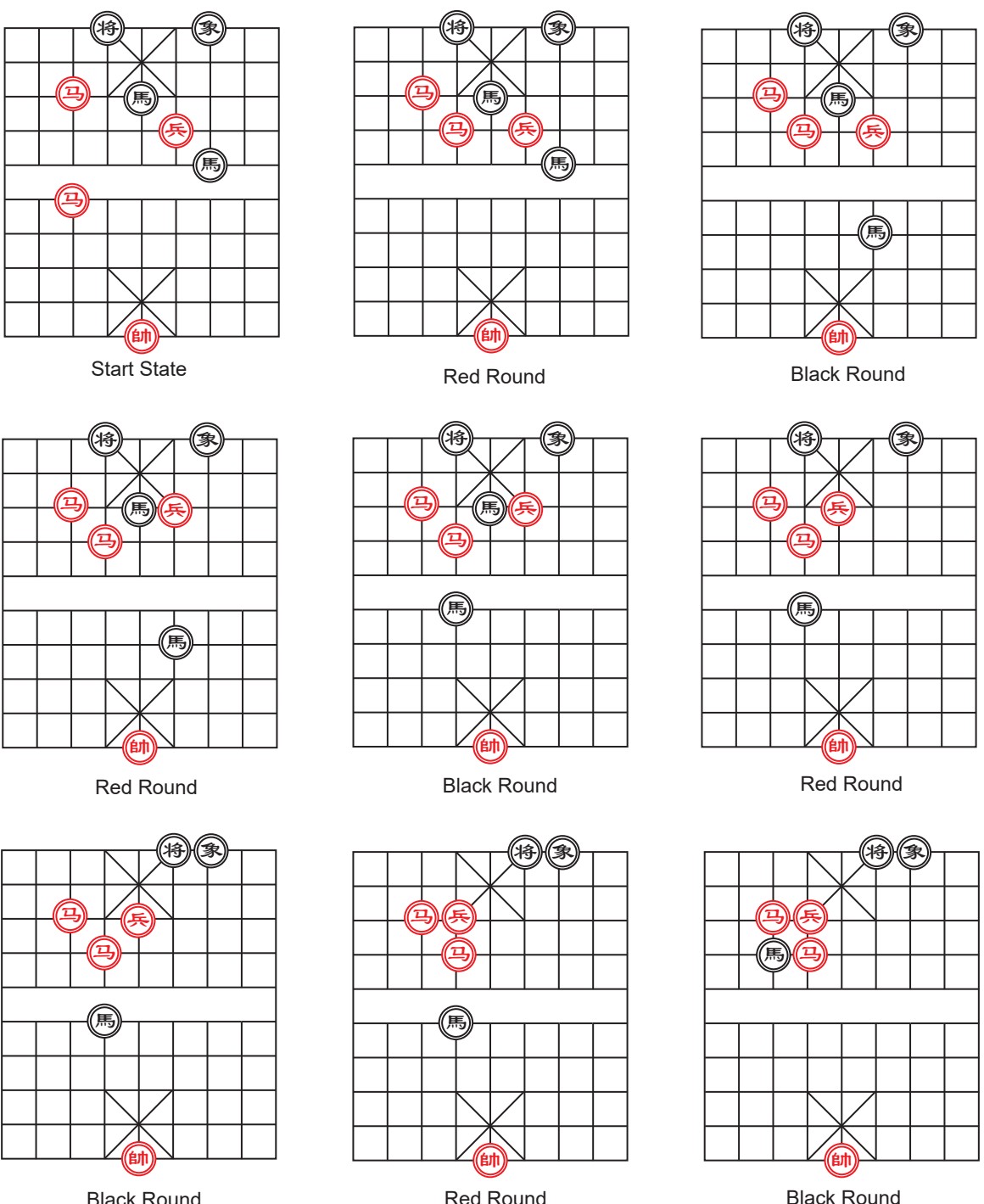

Figure 14: Game Trajectories of JiangJun playing endgame "double Knight and Pawn V.S. double Knight and single Bishop"

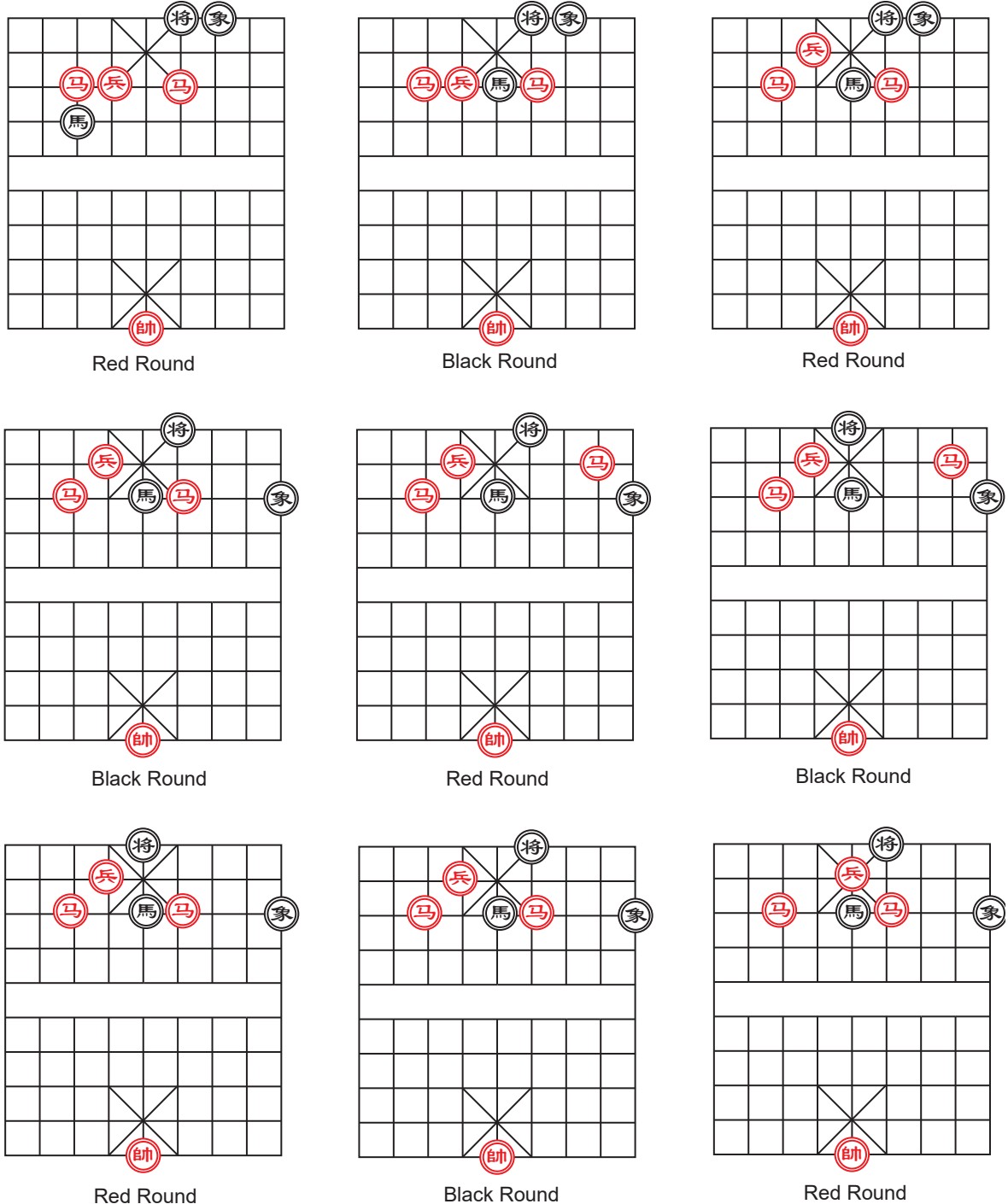

Figure 15: Game Trajectories of JiangJun playing endgame "double Knight and Pawn V.S. double Knight and single Bishop"

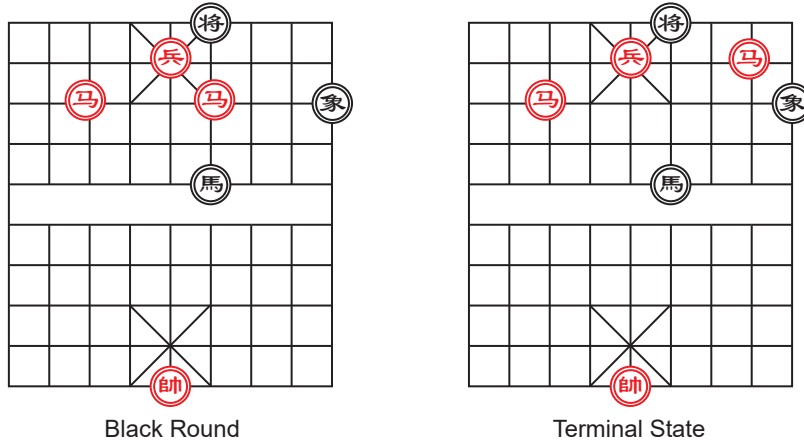

Figure 16: Game Trajectories of JiangJun playing endgame "double Knight and Pawn V.S. double Knight and single Bishop"

