# OpenReview forum: "JiangJun: Mastering Xiangqi by Tackling Non-Transitivity in Two-Player Zero-Sum Games"
_TMLR — Accepted by TMLR_

### Review · Reviewer_erju · 2023-04-05

**Summary Of Contributions:**

This paper considers the game of Xiangqi, a Chinese variant of Chess. The main contributions are: (1) A strong game AI called JiangJun for Xiangqi, trained by Monte-Carlo Tree-Search and Policy-Space Response Oracles, which achieves high win rates against human players and other algorithms such as AlphaZero; (2) Analyses of game plays in Xiangqi by both human and AI players, such as the game geometry (the landscape of the non-transitivity), and behaviors in challenging endgames.

**Audience:**

Yes

**Claims And Evidence:**

Yes

**Requested Changes:**

* Could the authors discuss in more details how different / similar is the JiangJun algorithm from existing algorithms such as AlphaZero or PSRO?

* The notation in Definition 4.1 & 4.2 are a bit confusing and lacks rigorous mathematical definition, it may be good to consider expanding those.

* I did not find a formal definition of  “transitivity” in the paper? It may be good to add one.


**Strengths And Weaknesses:**

Strengths:

* A main merit of the paper is the game Xiangqi, which has a large number of players in the population yet has not received much attention in the literature. Developing a strong game AI and understanding of this game could be of interest to the community.

* The trained AI agent achieves high win rates against both human players and existing training algorithms such as AlphaZero.

* The paper contains a substantial amount of engineering details, in addition to the results. The details could be useful for future work in other large games beyond Xiangqi.

* The analysis of the game plays is interesting, and could serve as a useful reference point for future studies on this game.

Weaknesses:

* The training algorithm seems like direct combinations of existing algorithms (MCTS, PSRO). This is perhaps expected and a minor concern given that the game itself is the main point of the paper.

* The presentation could be improved at several places (see the “Requested Changes” for details).

Overall, I am positive about the paper given its substantial contribution to the game Xiangqi. The paper is particularly suitable for TMLR, as the results are sound and could serve as a good reference point for future work.

---

> ### Author Response · Authors · 2023-05-09
>
> Thank you for taking the time to review our manuscript. We first appreciate your thorough assessment and helpful feedback on our paper, which we believe have helped us to significantly improve the quality of our work. In response to your feedback, please find below a point-by-point response to the comments. Furthermore, we will ensure that the appropriate modifications are incorporated into our revised manuscript.
>
> **Q: Could the authors discuss in more details how different / similar is the JiangJun algorithm from existing algorithms such as AlphaZero or PSRO?**
>
> **Response:** We are sorry for any confuses caused by this. The JiangJun algorithm, a novel approach that, to the best of our knowledge, is the first to synergistically combine MCTS and PSRO in order to tackle the complex issue of non-transitivity in Xiangqi. The algorithn is one of three main contributions in our paper. Other two contributions are 1) Through the analysis of over 10,000 human game records, we unveil the presence of non-transitivity in the Xiangqi game; 2) We establish a genuine human evaluation platform and carry out a series of experiments to validate our findings.
>
> **Q: The notation in Definition 4.1 & 4.2 are a bit confusing and lacks rigorous mathematical definition, it may be good to consider expanding those.**
>
> **Response:** We appreciate your insightful recommendations for enhancing the rigor and readability of our paper. We are sorry for the confusing caused by the Definitions 4.1 and 4.2, which are initially proposed by paper [1]. Here, we have revised definitions 4.1 and 4.2, and included additional clarifying explanations to ensure they are more comprehensible and straightforward. These updated definitions will be incorporated into our subsequent manuscript.
>
> **Definition 4.1** [k-layered finite Game of Skill [1]] In a game, if we can divide the set of strategies $\Pi$ into $k$ separate layers $L_i$ such that the union of all layers equals the set of strategies ($\cup_i L_i=\Pi$), and no two different layers share any strategies ($L_i\cap L_j=\emptyset$ for $i\neq j$), then the layers are fully transitive if the following conditions are satisfied: for all $i<j$ with $\pi_i \in \mathrm{L}_i$ and $\pi_j \in \mathrm{L}_j$, we have $\mathrm{f}\left(\pi_i, \pi_j\right)>0$, and there exists a number $z\in \mathbb{R}$ for which, if we compare the size of consecutive layers, we find that the layers grow in size up to layer $z$ ($|L_i|\leq |L_i+1|$ for each $i < z$) and then decrease in size for layers beyond $z$ ($|L_i|\geq|L_i+1|$ for each $i \geq z$).
>
> In simpler terms, the k-layered finite Game of Skill states that if a game's strategies can be organized into distinct layers that don't share strategies and satisfy a transitivity condition based on a function $\mathrm{f}$, then these layers are fully transitive. Additionally, there exists a threshold value $z$ that determines the point at which the layers switch from growing to decreasing in size.
>
> In practice, non-transitive interactions don't always follow a simple layer structure. To tackle this, Nash clustering was proposed, which relaxes transitive relations and creates a new cluster structure. This method finds the mixed Nash equilibrium of the game payoff $\mathcal{M}$ over the set of pure strategies $\Pi$ and forms clusters using the support of these mixtures. The process is repeated with remaining strategies until none are left.
>
> **Definition 4.2**[Nash Clustering [1]] In a finite two-player zero-sum symmetric game with a strategy set $\Pi$, Nash clustering is defined as $C:=(N_j:j\in \mathbb{N} \bigwedge N_j\neq \emptyset)$. For each integer $i \geq 1$, we have $N_{i+1}=supp(Nash(\mathcal{M}|\Pi \backslash \bigcup_{j\leq i} N_j))$ for $N_0=\emptyset$. Here, $supp$ denotes the support of the mixed strategy, and $Nash(\mathcal{M} | X)$ is the equilibrium for payoff $\mathcal{M}$ when restricted only to strategies in $X\in \Pi$.
>
> In simpler terms, Nash Clustering is a method used to identify mixed Nash equilibria in a game by considering the strategies not yet included in previous clusters. This approach allows for a more flexible understanding of non-transitive interactions, which may not be easily organized into a simple layer structure.

---

> ### Author Response · Authors · 2023-05-09
> **Response to Reviewer**
>
> **Q: I did not find a formal definition of “transitivity” in the paper? It may be good to add one.**
>
> **Response:** Thank you for pointing out the omission of a formal definition for "transitivity" in our paper. By the definition by [2], a game is transitive if there is a "rating function" $f$ such that performance on the game is the difference in ratings: $\phi(v, w)=f(v) - f(w)$. On the other hand, nontransitive games are characterized by strategic cycles, such as the well-known rock-paper-scissors game. We provide the following formal definition of nontransitivity, adapted from [2]: A game is considered nontransitive if: $\int_W \phi(\mathbf{v}, \mathbf{w}) \cdot d \mathbf{w}=0 \quad \text { for all } \quad \mathbf{v} \in W,$where $W$ denotes a set of agents, which can be parameterized by, for example, the weights of a neural network. $\phi$ represents an antisymmetric function that evaluates pairs of agents, with the domain and range defined as $\phi: W \times W \rightarrow \mathbb{R}$. We will add formal definitons of both "transitivity" and "intransitivity" in our further manuscript.
>
> **Reference:**
>
> [1] Wojciech Marian Czarnecki, Gauthier Gidel, Brendan Tracey, Karl Tuyls, Shayegan Omidshafiei, David Balduzzi, and Max Jaderberg. Real world games look like spinning tops, 2020.
>
> [2] Balduzzi, David, Marta Garnelo, Yoram Bachrach, Wojciech M. Czarnecki, Julien Pérolat, Max Jaderberg and Thore Graepel. “Open-ended Learning in Symmetric Zero-sum Games.” International Conference on Machine Learning (2019).

---

### Review · Reviewer_Dvv8 · 2023-04-18

**Summary Of Contributions:**

This paper considers building agents for a traditional board game called Xiangqi that is popular in China, and has a medium game tree complexity, akin to Chess and Shogi, based on a relatively small human dataset of 10,000 records. The authors show that the game structure has both transitive and intransitive components (akin to previous games like Starcraft, Blotto and others), and propose an algorithm that combines MCTS with PSRO to address this challenging surface. The empirical analysis shows that the method achieves a master level with a winrate over 99% over human players, and also includes population performance analysis and visualization results indicating how it deals with the non-transitivity of strategies.




**Audience:**

Yes

**Broader Impact Concerns:**

I do not see broader impact concerns

**Claims And Evidence:**

Yes

**Requested Changes:**

Figure 3 of the method itself is very helpful, and I’d suggest making a more detailed version and possibly moving it earlier in the paper. Please point in the figure itself to the relevant algorithms to make it easier for the reader. The algorithm, appendix D, belongs in the main text and not the appendices, I think.


I think it’d be good to have a discussion of the computational complexity / runtime (using O() notation) - while this is relatively simple, this would be useful to include. Similarly, a discussion of the runtime / complexity of solving for a Nash / MaxEnt Nash should be discussed (this is straightforward here as it is a two player zero sum game, more on this later).


I think an experimental analysis of what happens with a standard RL algorithm on this game is also welcome; I fully expect independent RL to do a lousy job due to cycles, but an empirical evaluation with multiple RL algorithm baselines could really highlight this.


In terms of existing work, a more detailed discussion of what methods have been attempted that combine Tree- Search or Reinforcement Learning and Game Theory / Nash solving would be welcome. You discuss the PSRO line of work - what attempts have been done to combine it with tree search? A table of existing methods and a description of what they use, in the style of the one on page 2 would be very welcome (e.g. population method, using RL vs Tree-Search, computing Nash equilibria, two player vs multiplayer, zero-sum vs general sum - etc.).


I think the paper clearly demonstrates the value of such approaches for the restricted class of two player zero sum games. However, I suspect the algorithm could be applicable to general sum games / mixed motive games, and many player games (i.e. games with *more* than two players). One key such domain is the game of (No-Press) Diplomacy, which would be good to mention - see the following:


Bakhtin, Anton, et al. "No-press diplomacy from scratch." Advances in Neural Information Processing Systems 34 (2021): 18063-18074.


Anthony, Thomas, et al. "Learning to play no-press diplomacy with best response policy iteration." Advances in Neural Information Processing Systems 33 (2020): 17987-18003.


(the last paper uses a Nash league analysis of agents to consider whether there are transitive and intrasitive components, which is a bit akin to your approach. This is also somewhat similar to the Starcraft league which you already discuss)


Diplomacy also has versions with agent communication, which might be an interesting usecase for such algorithms, see e.g.


Kramár, János, et al. "Negotiation and honesty in artificial intelligence methods for the board game of Diplomacy." Nature Communications 13.1 (2022): 7214.


Meta Fundamental AI Research Diplomacy Team (FAIR)†, et al. "Human-level play in the game of Diplomacy by combining language models with strategic reasoning." Science 378.6624 (2022): 1067-1074.


Similarly, one might consider other games, such as team formation and social or alliance dilemmas:


Baker, Bowen. "Emergent reciprocity and team formation from randomized uncertain social preferences." Advances in Neural Information Processing Systems 33 (2020): 15786-15799.


Hughes, Edward, et al. "Learning to resolve alliance dilemmas in many-player zero-sum games." arXiv preprint arXiv:2003.00799 (2020).


Leibo, Joel Z., et al. "Multi-agent reinforcement learning in sequential social dilemmas." arXiv preprint arXiv:1702.03037 (2017).


I think a section / appendix discussing what happens if one were to try these approaches in settings that are not completely zero sum or have more than two players is important. I see a couple of issues that might arise. First, computing the Nash equilibrium might become more computationally expensive; for 2 player zero sum game there is a simple linear program for solving for a Nash, which is not the case for games that have more players or games that are mixed motive (i.e. not zero sum, so agents may have partially aligned incentives). Second, convergence of best response dynamics (such as fictitious play) into a Nash is no longer guaranteed. This means the method might do a really bad job in generating agents for such games (in other words, some dynamics just iterate forever on the intransitive cycles). I’d love to see your conjectures on how the method might perform in such settings (you might even try it out on a simple game, such as e.g. an auction?).


There are some discussions of such topics in various surveys / gyms, e.g.:


Lv, Yongfeng, and Xuemei Ren. "Approximate Nash solutions for multiplayer mixed-zero-sum game with reinforcement learning." IEEE Transactions on Systems, Man, and Cybernetics: Systems 49.12 (2018): 2739-2750.


ning and control (2021): 321-384.Leibo, Joel Z., et al. "Scalable evaluation of multi-agent reinforcement learning with melting pot." International conference on machine learning. PMLR, 2021.


(this one might have some environments to try your algorithm on)


Dafoe, Allan, et al. "Open problems in cooperative AI." arXiv preprint arXiv:2012.08630 (2020).


Zhang, Kaiqing, Zhuoran Yang, and Tamer Başar. "Multi-agent reinforcement learning: A selective overview of theories and algorithms." Handbook of reinforcement lear




To conclude, this is an exciting piece of research, with a great demonstration of combining tree search and PSRO / double oracle agent construction, showing human level performance in an interesting game. Most of my comments are about presentation and showing the right context of existing work.



**Strengths And Weaknesses:**

Overall, I find the paper exciting and the results convincing. In particular, I’m excited with how the authors combine game theoretic principles and search / MCTS techniques to develop a state of the art agent. The combination of a population based approach with game theoretic algorithms and computing Nash equilibria is very compelling, and the empirical analysis is convincing. While the game of Xiangqi itself was new to me, the discussion of the game is sufficient to understand why it is challenging and interesting.


Most of my comments relate to the presentation and clarity of the paper, and issues regarding placing it appropriately in the context of existing work.

---

> ### Author Response · Authors · 2023-05-09
> **Response to Reviewer**
>
> We sincerely thank you for the thorough evaluation of our manuscript. Your constructive feedback has helped us improve our work, and we will incorporated the necessary revisions into our manuscript.
>
> **Q: Figure 3 of the method itself is very helpful, and I’d suggest making a more detailed version and possibly moving it earlier in the paper. Please point in the figure itself to the relevant algorithms to make it easier for the reader. The algorithm, appendix D, belongs in the main text and not the appendices, I think.**
>
> **Response:** Thank you for your insightful recommendations. We apologize for our inability to upload images on the OpenReview system. We assure you that we will enhance Figure 3 by providing more details, and relocate the algorithm from Appendix D to Section 5, titled "JiangJun: The Method.
>
> **Q: I think it’d be good to have a discussion of the computational complexity / runtime (using O() notation) ...**
>
> **Response:** We appreciate your valuable feedback and will incorporate a discussion on computational complexity in our manuscript. We will focus our analysis on the two primary components of JiangJun: the MCTS Actor and the Nash Solver. The worst-case time complexity of the MCTS Actor can be denoted as $O(b^d * n * C)$, where $b$ signifies the effective branching factor, $d$ represents the effective search depth, $n$ refers to the number of simulations, and $C$ corresponds to the inference time of the neural network.
>
> In Xiangqi, the average branching factor can vary between 30 and 80, depending on the specific position. The game tree depth for Xiangqi might extend to 40 or more. The number of simulation times $n$ is typically set at 800 or 1800. The inference time of the neural network $C$ is influenced by factors such as network size and architecture, implementation efficiency, and the employment of hardware accelerators like GPUs or TPUs.
>
> It is crucial to emphasize that the time complexity $O(b^d * n * C)$ represents a worst-case approximation, and the actual time complexity is often significantly lower than this estimate.
>
> Regarding the Nash Solver, we compute a unique maximum entropy Nash equilibrium by solving a Linear Programming (LP) problem, as described in Eq.2 of our manuscript. At iteration $n$, player 1 and player 2 each have $k$ strategies. The time complexity of solving an LP problem is contingent on the specific algorithm employed. For example, the worst-case time complexity of the Simplex method is $O(2^k * poly(k))$, where $poly(k)$ is a polynomial in k.

---

> ### Author Response · Authors · 2023-05-09
> **Response to Reviewer**
>
> **Q: I think an experimental analysis of what happens with a standard RL algorithm on this game is also welcome...**
>
> **Response:** Thanks for your advice. On the one hand, we have presented some empirical investigation of the non-transitivity issue arising in AlphaZero. In Section 4 and Figure 2, we have analyzed and demonstrated that the standard RL algorithm (AlphaZero) results in non-transitivity, as evidenced by the cyclical pattern among three trained strategies. Furthermore, Figure 6 displays the Nash equilibrium distributions of AlphaZero (on the left), which also exhibit the non-transitivity problem. Additionally, in our current manuscript version, we provide a comparison with the standard AlphaZero Xiangqi algorithm, which has achieved a strength comparable to a 9-dan player. We apologize for any unclear descriptions of the strength comparison in our current manuscript. As depicted in Figure 4 (a), the AlphaZero Xiangqi reached a win rate of about 60% against the AlphaZero Xiangqi method after the first 200k training steps. By the end of training, the AlphaZero Xiangqi achieved a win rate of about 85% after more than 700k training steps, with the win rate fluctuating during the last approximately 100k training steps. To enhance clarity, we plan to separate Figure 4 (a) into two parts: one figure illustrating the training progression of RP-ELO and one table presenting the win rates against standard AlphaZero, behavior cloning algorithms, and real human players. This will allow readers to more easily understand the strength levels attained by our model.
>
> On the other hand, we offer an additional strength evaluation by comparing our approach with two traditional RL methods: behavior cloning Xiangqi algorithm and standard AlphaZero Xiangqi algorithm. First, we include a comparison with the behavior cloning Xiangqi method, which was trained using a dataset of 300,000 expert Xiangqi player data samples. Our JiangJun model has demonstrated the ability to **defeat BC Xiangqi with a win rate of 96.40\%**, as determined from a sample of 111 games. The specifics of the behavior cloning model for Xiangqi are outlined below. To train the model, we collected and processed a dataset consisting of 300,000 Xiangqi data samples. Each sample is composed of an input-output pair $(s, a)$, where input $s$ represents the state as a $9 \times 10\times 14$ binary matrix, and output $a$ is a one-hot action vector with 2048 dimensions. The state representation is identical to that used in our JiangJun model, as described in detail in Appendix B. Regarding the state representation, each of the 14 planes is a $9 \times 10$ matrix, with the first seven planes representing the positions of the red player's pieces and the last seven planes representing the positions of the black player's pieces. We utilized the ResNet-18 architecture [1] as the basis for our behavior cloning approach. The ResNet-18 model takes the state as input and predicts the corresponding action. By training this model on our extensive dataset, our goal was to effectively capture the essence of human gameplay and decision-making in Xiangqi.

---

> ### Author Response · Authors · 2023-05-09
> **Response to Reviewer**
>
> **Q: In terms of existing work, a more detailed discussion of what methods have been attempted that combine Tree-Search or Reinforcement Learning and Game Theory / Nash solving would be welcome. You discuss the PSRO line of work - what attempts have been done to combine it with tree search? A table of existing methods and a description of what they use, in the style of the one on page 2 would be very welcome ...**
>
> **Response:** We are grateful for your insightful suggestions. To our knowledge, our research is the first to effectively merge MCTS and PSRO. However, there is a wealth of existing work on the integration of MCTS and RL. We have prepared a more comprehensive discussion on this topic, organized by year, which we plan to present as a timeline chart (unfortunately, we are unable to upload figures in the OpenReviewer system). The next paragraph discussing the combination of MCTS and RL refers to references [2, 13]. We will include the timeline chart and subsequent discussion in our updated manuscript.
>
> MCTS, initially a standalone algorithm, was connected with RL from Silver's 2009 Ph.D. thesis[1], according to Vodopivec et al.[2]. The TD-MCTS approach[3], introduced in 2014, incorporated TD learning into MCTS, replacing the UCT formula and modifying state estimate calculations for nodes. The fusion of MCTS and RL gained breakthrough with the advent of AlphaGo[4] in late 2015. In 2017, MoHex-CNN[5] employed the AlphaGo strategy to excel at the game of Hex,  while AlphaGoZero was proposed to master Go without human knowledge [14]. DeepMind's AlphaZero [6] was then proposed and excelled at chess, shogi, and Go. In the same year, Ilhan et al. devised a technique using the TD method to adjust the policy during MCTS's simulation phase[11]. MoHex-3HNN[7], with its groundbreaking three-head neural network architecture, significantly outperformed MoHex-CNN in 2018. Soemers et al.[8] in 2019, learned a policy in an MPD using the policy gradient method and value estimates directly from MCTS. MCTS also acted as a demonstrator for the RL component in [9]. In 2020, a method inspired by AlphaGo was created to operate without prior knowledge of komi[10]. In 2022, Scheiermann et al. integrated MCTS with TD n-tuple networks for the first time, using this combination only during testing to create adaptable agents while maintaining low computational demands[12].
>
> **Reference**
> [1] Silver, D. (2009). Reinforcement learning and simulation-based search in computer Go. Ph.D. thesis, University of Alberta, Edmonton, Alta., Canada.
>
> [2] Vodopivec, Tom, Spyridon Samothrakis, and Branko Ster. "On monte carlo tree search and reinforcement learning." Journal of Artificial Intelligence Research 60 (2017): 881-936.
>
> [3] Vodopivec T, Šter B (2014) Enhancing upper confidence bounds for trees with temporal difference values. In: 2014 IEEE conference on computational intelligence and games. IEEE, pp 1–8
>
> [4] Silver D, Huang A, Maddison CJ, Guez A, Sifre L, Van Den Driessche G, Schrittwieser J, Antonoglou I, Panneershelvam V, Lanctot M et al (2016) Mastering the Game of Go with deep neural networks and tree search. Nature 529(7587):484–489
>
> [5] Gao C, Hayward R, Müller M (2017) Move prediction using deep convolutional neural networks in Hex. IEEE Trans Games 10(4):336–343
>
> [6] Silver, David, Thomas Hubert, Julian Schrittwieser, Ioannis Antonoglou, Matthew Lai, Arthur Guez, Marc Lanctot et al. "Mastering chess and shogi by self-play with a general reinforcement learning algorithm." arXiv preprint arXiv:1712.01815 (2017).
>
> [7] Gao C, Takada K, Hayward R (2019) Hex 2018: MoHex3HNN over DeepEzo. J Int Comput Games Assoc 41(1):39–42
>
> [8] Soemers DJ, Piette E, Stephenson M, Browne C (2019) Learning policies from self-play with policy gradients and MCTS value estimates. In: 2019 IEEE conference on games (CoG). IEEE, pp 1–8
>
> [9] Kartal B, Hernandez-Leal P, Taylor ME (2019) Action guidance with MCTS for deep reinforcement learning. In: Proceedings of the AAAI conference on artificial intelligence and interactive digital entertainment, vol 15, pp 153–159
>
> [10] Yang B, Wang L, Lu H, Yang Y (2020) Learning the Game of Go by scalable network without prior knowledge of Komi. IEEE Trans Games 12(2):187–198
>
> [11] Ilhan E, Etaner-Uyar AŞ (2017) Monte Carlo Tree Search with temporal-difference learning for General Video Game Playing. In: 2017 IEEE conference on computational intelligence and games (CIG). IEEE, pp 317–324
>
> [12] Scheiermann, Johannes, and Wolfgang Konen. "AlphaZero-Inspired General Board Game Learning and Playing." arXiv preprint arXiv:2204.13307 (2022).
>
> [13] Świechowski, Maciej, Konrad Godlewski, Bartosz Sawicki, and Jacek Mańdziuk. "Monte Carlo tree search: A review of recent modifications and applications." Artificial Intelligence Review 56, no. 3 (2023): 2497-2562.
>
> [14] Silver, D., Schrittwieser, J., Simonyan, K. et al. Mastering the game of Go without human knowledge. Nature 550, 354–359 (2017).

---

> ### Author Response · Authors · 2023-05-09
> **Response to Reviewer**
>
> **Q: I suspect the algorithm could be applicable to general sum games / mixed motive games, and many player games ... I think a section / appendix discussing what happens if one were to try these approaches in settings that are not completely zero sum or have more than two players is important.**
>
> **Response:** We appreciate your valuable suggestion and the list of related references. The primary focus of this paper is to investigate and address the non-transitivity problem inherent in the Xiangqi game. As a result, our research is centered on zero-sum, two-player competitive games. However, we recognize the importance of considering broader game settings, such as general sum games, mixed-motive games, and games with multiple players. Your input is indeed valuable, and we plan to explore these aspects in our future work.

---

### Review · Reviewer_vM5P · 2023-04-28

**Summary Of Contributions:**

In this paper, the authors do the following:

1. They analyse a dataset of >10 000 online games, and train three AlphaZero agents to play Xiangqi. They use this to conclusively demonstrate that Xiangqi is a game with non-transitivity.
2. They propose a novel method, the JiangJun algorithm, to train an algorithm to deal with this non-transitivity.
3. They use their algorithm to train an agent and evaluate it by playing against humans, playing >7000 games and winning >97% of them.

The authors do a number of excellent experiments. However, these experiments are presented in a confusing manner.



**Audience:**

Yes

**Broader Impact Concerns:**

I have no ethical concerns about this paper.

**Claims And Evidence:**

Yes

**Requested Changes:**

Proposed adjustments to the submission:

1. Either head-to-head evaluations against comparable algorithms, or some work to assess the exploitability of the agent (critical).
2. A more clear strength evaluation (strengthen).
3. A clearer description of the evaluation against humans (critical).
4. A clearer understanding of how the same non-transitivity analysis does in other games (strengthen).
5. A clearer description of the contributions made by this paper, and a clear demarcation of what comes from the existing literature (critical).
6. Additional copy-editing to fix the typos (strengthen).

**Strengths And Weaknesses:**

The problems that arise in Xiangqi are very similar to the problems that arise in the imperfect information games literature. In particular, finding a Nash Equilibrium, as they do here, addresses the transitivity issue. There are a large number of algorithms from the imperfect information literature which they could use as baselines. Additionally, there are a number of standard metrics, such as exploitability and exploitability approximations, which would help indicate the strength of their agent.

Adding these evaluations would help readers place the JiangJun agent in context and better understand the strength of their proposal.

Additionally, the paper could engage more with the existing imperfect information literature. It would be very nice to see the authors evaluate against the Player of Games algorithm, or Libratus. It would be nice to evaluate against some of these agents in comparable domains.

The evaluation against humans, while nice to have, was also confusing. For one, the authors have two stages, “developing” and “done”. I think it would be more natural to refer to this as “training” and “evaluation”, and to provide the number of training steps at each point in the process. It seems surprising that the agents only improve from 97.42% to 99.35% win-rates against humans. That indicates, to me, that the humans are not particularly good at Xiangqi.

Additionally, it is unclear to me why the last 3 months are broken out separately if the agent is not being trained further. The win rate per month is irrelevant, as what we care about is the overall win-rate.

For the non-transitivity analysis, the authors train three AlphaZero agents, which were generated sequentially by self-play. They demonstrate that the agents exhibit non-transitivity: black wins against red >50% of the time, red wins against blue >50% of the time, but red wins against black >50% of the time.

To me, this seems like an ideal scenario to measure exploitability (or an approximation thereof). What I did not understand was how surprising is this? If one were to follow a similar process in, say, Chess, or Poker, would one expect similar results?

In addition to the issues evaluating the strength of the agent, I found it difficult to understand the contributions that the authors are claiming. For instance, a significant amount of the paper (most of the “MCTS Actor” and “training” sections) are simply restating the AlphaZero algorithm. It would be great to see an explicit breakdown of the contributions that this paper makes, and where they depart from a vanilla reimplementation of AlphaZero.

I would encourage the authors to rely more on their citations. For instance, they could spend less time describing AlphaZero and simply cite that paper.

Finally, there are a number of typos in the paper. This wasn’t a major factor in my decision, but the paper would benefit from copy-editing. For instance, “At 940k steps, JiangJun agent can win 99.35% human players in our JiangJun mini-program.” I believe this should read “At 940k steps, JiangJun agent can *beat* 99.35% *of the* human players in our JiangJun mini-program.”

In short, this paper involves a number of strong experiments and proposes an interesting algorithm, but would benefit from additional editing and exposition to make the contributions more clear.

---

> ### Author Response · Authors · 2023-05-09
> **Response to Reviewer**
>
> We would like to express our gratitude for your time and effort in reviewing our manuscript. We appreciate your valuable feedback and have carefully addressed each comment in our revised manuscript. Please find below our point-by-point responses to the reviewers' comments. And we promise that we will incorporated the necessary revisions into our manuscript.
>
> **Q: Either head-to-head evaluations against comparable algorithms, or some work to assess the exploitability of the agent. And a more clear strength evaluation.**
>
> **Response:** We appreciate your valuable feedback regarding the potential study of exploitability and exploitability approximations in Xiangqi. While there is a substantial body of work on exploitability approximations, we acknowledge that studying Xiangqi in this context presents unique challenges. Consequently, we believe it would be best to address this topic as a separate, independent research project in the future.
>
> On the other hand, we offer an additional strength evaluation by comparing our approach with two traditional RL methods: behavior cloning Xiangqi algorithm and standard AlphaZero Xiangqi algorithm. First, we include a comparison with the behavior cloning Xiangqi method, which was trained using a dataset of 300,000 expert Xiangqi player data samples. This comparison will help to contextualize our method's performance relative to other established RL techniques. Our JiangJun model has demonstrated the ability to **defeat BC Xiangqi with a win rate of 96.40\%**, as determined from a sample of more than 100 games. The specifics of the behavior cloning model for Xiangqi are outlined below. To train the model, we collected and processed a dataset consisting of 300,000 Xiangqi data samples. Each sample is composed of an input-output pair $(s, a)$, where input $s$ represents the state as a $9 \times 10\times 14$ binary matrix, and output $a$ is a one-hot action vector with 2048 dimensions. The state representation is identical to that used in our JiangJun model, as described in detail in Appendix B. Regarding the state representation, each of the 14 planes is a $9 \times 10$ matrix, with the first seven planes representing the positions of the red player's pieces and the last seven planes representing the positions of the black player's pieces. We utilized the ResNet-18 architecture [1] as the basis for our behavior cloning approach. The ResNet-18 model takes the state as input and predicts the corresponding action. By training this model on our extensive dataset, our goal was to effectively capture the essence of human gameplay and decision-making in Xiangqi.
>
> Additionally, in our current manuscript version, we provide a comparison with the standard AlphaZero Xiangqi algorithm, which has achieved a strength comparable to a 9-dan player. We apologize for any unclear descriptions of the strength comparison in our current manuscript. As depicted in Figure 4 (a), the AlphaZero Xiangqi reached a win rate of about 60% against the AlphaZero Xiangqi method after the first 200k training steps. By the end of training, the AlphaZero Xiangqi achieved a win rate of about 85% after more than 700k training steps, with the win rate fluctuating during the last approximately 100k training steps. The findings are further corroborated by the win rate table featuring human players in the subsequent response. In the final training month, JiangJun achieved a 99.35% win rate, while the evaluation win rate three months prior remained nearly identical at 99.41%. To enhance clarity, we plan to separate Figure 4 (a) into two parts: one figure illustrating the training progression of RP-ELO and one table presenting the win rates against standard AlphaZero, behavior cloning algorithms, and real human players. This will allow readers to more easily understand the strength levels attained by our model.
>
> Thank you once more for your valuable suggestions. We will incorporate the aforementioned discussion into our future manuscript revisions.
>
> [1] He, Kaiming, Xiangyu Zhang, Shaoqing Ren, and Jian Sun. "Deep residual learning for image recognition." In Proceedings of the IEEE conference on computer vision and pattern recognition, pp. 770-778. 2016.

---

> ### Author Response · Authors · 2023-05-09
> **Response to Reviewer**
>
> **Q: A clearer description of the evaluation against humans.**
>
> **Response:** We apologize for the confusion caused by Table 2 in our manuscript, which describes the results against human players. Firstly, we agree that "training" and "evaluation" are more appropriate terms and will ensure they are used correctly in the revised manuscript. Next, we are sorry for missing important information that the release date of the JiangJun-human playing platform does not coincide with the model's training time, as the platform was launched approximately one month later. Consequently, the JiangJun model, which achieved a 97.42% win rate against human players in the first month, is not a model with random weights. Additionally, we update the model weights on the platform weekly. Therefore, the improvement from 97.42% to 99.35% is not from the beginning of training.
>
> We appreciate your suggestion regarding the representation of data. We will consolidate the data from the last three months and report the overall win rate. The revised table, as shown below, will be included in the updated manuscript. it is worth noting that the final win rate of 99.41 presented in the table is not consistent with the 99.39% in our manuscript. The discrepancy arose from calculating the final win rate by averaging the rounded win rates over the three months in our manuscript.
>
> | **Month** | **Stage** | **Wins** | **Ties** | **Losses** | **Total** | **Win Rate** |
> |:------------------:|:------------------:|:-----------------:|:-----------------:|:-------------------:|:------------------:|:---------------------:|
> | **Month 1**        | Training           | 717               | 11                | 8                   | 736                | 97.42\%               |
> | **Month 2**        | Training           | 724               | 0                 | 17                  | 741                | 97.71\%               |
> | **Month 3**        | Training           | 462               | 0                 | 3                   | 465                | 99.35\%               |
> | **Month 4-6**      | Evaluation         | 5089              | 3                 | 27                  | 5119               | 99.41\%               |
>
> We will ensure that all data is accurately updated in the revised manuscript.
>
> **Q: A clearer understanding of how the same non-transitivity analysis does in other games.**
>
> **Response:** Thank you for your insightful suggestions aimed at improving the clarity and readability of our paper. We appreciate your input and assure you that we will include the following discussion on how non-transitivity analysis applies to other games in the revised version of the paper. Empirical analyses of various real-world games have demonstrated the existence of non-transitivity issues, spanning from traditional board games to modern video games. As indicated in paper [1], games such as Connect Four, Tic Tac Toe, Misere Tic Tac Toe (a Tic Tac Toe variant where a player wins if and only if their opponent forms a line), Hex $3\times3$, Go $3\times3$, Go $4\times4$, Quoridor $3\times3$, Quoridor $4\times4$, and StarCraft II (AlphaStar [2]) all display non-transitivity problems. In the case of StarCraft II, the authors analyzed the payoff matrix of the League of the AlphaStar Final [3], which represents a population of 900 agents with diverse skill levels. In our paper, we primarily focus on examining the fully complex board game Xiangqi using human player records, as opposed to merely studying simplified in paper [1], small-board versions of games (such as Go $4\times4$).
>
> [1] Czarnecki, Wojciech M., Gauthier Gidel, Brendan Tracey, Karl Tuyls, Shayegan Omidshafiei, David Balduzzi, and Max Jaderberg. "Real world games look like spinning tops." Advances in Neural Information Processing Systems 33 (2020): 17443-17454.
>
> [2] Oriol Vinyals, Igor Babuschkin, Wojciech M Czarnecki, Michaël Mathieu, Andrew Dudzik, Junyoung Chung, David H Choi, Richard Powell, Timo Ewalds, Petko Georgiev, et al. Grandmaster level in starcraft ii using multi-agent reinforcement learning. Nature, 575(7782):350–354, 2019.
>
> [3] Oriol Vinyals, Igor Babuschkin, Wojciech M Czarnecki, Michaël Mathieu, Andrew Dudzik, Junyoung Chung, David H Choi, Richard Powell, Timo Ewalds, Petko Georgiev, et al. Grandmaster level in starcraft ii using multi-agent reinforcement learning. Nature, 575(7782):350–354, 2019.

---

> ### Author Response · Authors · 2023-05-09
> **Response to Reviewer**
>
> **Q: A clearer description of the contributions made by this paper, and a clear demarcation of what comes from the existing literature.**
>
> **Response:** We appreciate your valuable feedback and apologize for any ambiguity in the presentation of our contributions. Our work offers three primary contributions: 1) Through the analysis of over 10,000 human game records, we unveil the presence of non-transitivity in the Xiangqi game; 2) We introduce the JiangJun algorithm, a novel approach that, to the best of our knowledge, is the first to synergistically combine MCTS and PSRO in order to tackle the complex issue of non-transitivity in Xiangqi; 3) We establish a genuine human evaluation platform and carry out a series of experiments to validate our findings.
>
> Although MCTS has been widely used in well-known algorithms like AlphaGo and AlphaZero, we recognize the importance of providing a detailed description of the methodology in our paper. This will help readers who may not be familiar with this field to better understand our approach and also serve as a useful reminder for those who are already acquainted with it. We appreciate your suggestion regarding demarcating our work from the existing literature, and we will make sure to do so in a more prominent location. Additionally, in this paper, our primary focus is on the perfect information game, Xiangqi. As a result, we apologize for not engaging with a more extensive range of imperfect information literature, except for those related to PSRO in imperfect games, such as Starcraft II.
>
> **Q: Additional copy-editing to fix the typos.**
>
> **Response:** Thank you for your thorough review and valuable feedback on our paper. We sincerely apologize for the typographical errors that led to confusion in the text. For example, the wrong sentence you mentioned is describing the left-upper sub-figure in Figure 2, which should be "black wins against red >50% of the time, red wins against blue >50% of the time, but blue wins against black >50% of the time", as shown in the figure. We deeply regret any inconvenience caused by these errors and assure you that we will rectify them in the revised version of the paper.

---

### Decision · Action_Editors · 2023-06-22

**Recommendation:** Accept with minor revision

**Comment:**

We are suggesting accept under revision asking the authors to make the strength comparison a lot stronger which should give relatively little work on their part, and we think it would make the paper a lot stronger. This should include computing the exploitability of the trained agents and others. All reviewers agree that the strength assessment as detailed in the review from vM5P is crucial.

**Audience:**

Xiangqi is a popular real-world game of reasonably large scale, and this is perhaps the first dedicated game AI paper on it.

**Claims And Evidence:**

They analyse a dataset of >10 000 online games, and train three AlphaZero agents to play Xiangqi. They use this to conclusively demonstrate that Xiangqi is a game with non-transitivity. They propose a novel method, the JiangJun algorithm, to train an algorithm to deal with this non-transitivity. They use their algorithm to train an agent and evaluate it by playing against humans, playing >7000 games and winning >97% of them.